**Magma ascent mechanisms in the transition regime from solitary porosity waves to diapirism**

Janik Dohmen[1], Harro Schmeling[1]

[1]Institute for Geoscience, Goethe University, Frankfurt, Germany

*Correspondence to:* dohmen@geophysik.uni-frankfurt.de

**Abstract**

In partially molten regions inside the Earth melt buoyancy may trigger upwelling of both solid and fluid phases, i.e. diapirism. If the melt is allowed to move separately with respect to the matrix, melt perturbations may evolve into solitary porosity waves. While diapirs may form on a wide range of scales, porosity waves are restricted to sizes of a few times the compaction length. Thus, the size of a partially molten perturbation in terms of compaction length controls whether material is dominantly transported by porosity waves or by diapirism. We study the transition from diapiric rise to solitary porosity waves by solving the two-phase flow equations of conservation of mass and momentum in 2D with porosity dependent matrix viscosity. We systematically vary the initial size of a porosity perturbation from 1.8 to 120 times the compaction length.

If the perturbation is of the order of a few compaction lengths, a single solitary wave will emerge, either with a positive or negative vertical matrix flux. If melt is not allowed to move separately to the matrix a diapir will emerge. In between these end members we observe a regime where the partially molten perturbation will split up into numerous solitary waves, whose phase velocity is so low compared to the Stokes velocity that the whole swarm of waves will ascend jointly as a diapir, just slowly elongating due to a higher amplitude main solitary wave.

Only if the melt is not allowed to move separately to the matrix no solitary waves will build up, but as soon as two-phase flow is enabled solitary waves will eventually emerge. The required time to build them up increases non-linearly with the perturbation radius in terms of compaction length and might be for many cases too long to allow for them in nature.

**1 Introduction**

In geodynamic settings such as mid-ocean ridges, hotspots, subduction zones or orogenic belts partial melts are generated within the asthenosphere or lower continental crust and ascend by fluid migration within deforming rocks (e.g., Sparks and Parmentier, 1991; Katz, 2008; Keller et al., 2017; Schmeling et al., 2019). Inherent tectonic or rock heterogeneities in such systems may result in spatially varying melt fractions on length scales varying over several orders of magnitudes. These length scales play an important role in determining whether melt anomalies may rise as porous waves (Jordan et al., 2018) or by other mechanisms such as diapirs (Rabinowicz et al., 1987), focused channel networks (Spiegelman

et al., 2001) or dykes (Rivalta et al., 2015). Here we focus on the effect of the length scale on the formation and evolution of buoyancy driven porous waves or diapirs.

The physics of fluid moving relatively to a viscously deformable porous matrix were firstly described by McKenzie (1984) and it was later shown by several authors that these equations allow for the emergence of solitary porosity waves (Scott & Stevenson, 1984; Barcilon & Lovera 1989; Wiggins & Spiegelman, 1995). Porosity waves are regions of localized excess fluid that ascend with permanent shape and constant velocity, controlled by compaction and decompaction of the surrounding matrix.

They have extensively been studied as mechanisms transporting geochemical signatures or magma through the asthenosphere, lower and middle crust (e.g. Watson & Spiegelman, 1994; McKenzie, 1984; Connolly, 1997; Connolly & Podladchikov, 2013, Jordan et al., 2018, Richard et al., 2012). It has been shown that the dynamics of porous waves strongly depends on the porosity dependence of the matrix rheology (e.g. Connolly & Podladchikov, 1998; Yarushina et al., 2015; Connolly & Podladchikov, 2015;

Omlin et al., 2017; Dohmen et al., 2019). Yet, one open question is how the length scale of solitary porosity waves relates to an arbitrary length scale of a possible porosity anomaly in given geodynamic settings.

The size of a solitary porosity wave is usually of the order of a few compaction lengths (McKenzie, 1984; Scott & Stevenson, 1984; Simpson & Spiegelman, 2011), but this length scale varies over a few

orders of magnitude, depending on the shear and bulk viscosity of the matrix, fluid viscosity and permeability (see eq. (19)) with typical values of 100-10000 meters (McKenzie, 1984; Spiegelman, 1993). However, partially molten regions in the lower crust or upper mantle are prone to gravitational instabilities such as Rayleigh-Taylor instabilities or diapirism (e.g. Griffith, 1986; Bittner and Schmeling, 1995; Schmeling et al., 2019). Originating from the Greek "*diapeirein*", i.e. "to pierce

through", diapirism describes the "buoyant upwelling of relatively light rock" (Turcotte & Schubert, 1982) through and into a denser overburden. In the general definition the rheology of the diapir and ambient material is not specified, both can be ductile as in our case. Buoyancy may be of compositional or phase related origin, e.g., due to the presence of non-segregating partial melt (Wilson, 1989). In this model we describe a diapir as a partially molten perturbation, whose rising velocity, characterizable by

the Stokes velocity, is lower than the corresponding solitary waves phase velocity.

As characteristic wavelengths of Rayleigh-Taylor instabilities may be similar, but also of significantly different order of those of porosity waves, and the Stokes velocity is strongly affected by the spatial expansion, the question arises how these two mechanisms interact and how does the transition from a porosity wave to a rising partially molten diapir look like. Scott (1988) already investigated a similar

scenario. He calculated porosity waves changing the compaction length by altering the constant shear to bulk viscosity ratio. In contrast, we vary the radius of a partially molten perturbation in terms of compaction lengths but keeping the porosity dependent viscosity law the same. While Scott (1988) was

not able to reach the single-phase flow endmember due to his setup, we can reach this endmember with our description and can explore the transition.

In this work we will address the question of length scale of a partially molten region with respect to the length scale of a solitary porosity wave, by varying the sizes of initial porosity perturbations. We further focus on the numerical implications on modelling magma transport.

## 2 Methods

### 2.1 Governing equations

The formulation of the governing equations for the melt-in-solid two-phase flow dynamics is based on McKenzie (1984), Spiegelman & McKenzie (1987) and Schmeling (2000) assuming an infinite Prandtl number, a low fluid viscosity w.r.t. the effective matrix viscosity, zero surface tension, and the Boussinesq approximation. In the present formulation the Boussinesq approximation assumes the same constant density for the solid and fluid except for the buoyancy terms of the momentum equations for

the solid and fluid. In the following all variables associated with the pore fluid (melt) have the subscript $f$ and those associated with the solid matrix have the subscript $s$. The equation for the conservation of the mass of the melt is

$$\frac{\partial \varphi}{\partial t} + \vec{\nabla} \cdot \left( \varphi \vec{v}_f \right) = 0, \tag{1}$$

and the mass conservation of the solid is

$$\frac{\partial (1-\varphi)}{\partial t} + \vec{\nabla} \cdot ((1 - \varphi)\vec{v}_s) = 0. \tag{2}$$

$\varphi$ is the volumetric rock porosity (often called melt fraction), $\vec{v}_f$ and $\vec{v}_s$ are the fluid and solid velocities, respectively. The momentum equations are given as a generalized Darcy equation for the fluid separation flow

$$\vec{v}_f - \vec{v}_s = -\frac{k_\varphi}{\mu \varphi} \left( \vec{\nabla} P_f - \rho_f \vec{g} \right), \tag{3}$$

where $\rho_f$ is the fluid density and $P_f$ is the fluid pressure (including the lithostatic pressure), whose gradient is driving the fluid segregation by porous flow, μ is the melt dynamic viscosity and $\vec{g}$ is the gravitational acceleration. $k_\varphi$ is the permeability that depends on the rock porosity

$$k_\varphi = k_0 \varphi^n, \tag{4}$$

with $n$ being the power-law exponent constant, usually equal to 2 or 3. This relation is known as the

Kozeny-Carman relation (e.g. Costa, 2006). The Stokes equation for the mixture is given as

$$\bar{\rho}\vec{g} - \vec{\nabla} P_f + \vec{\nabla} \cdot \boldsymbol{\tau} = 0. \tag{5}$$

$\bar{\rho}$ is the density of the melt – solid mixture and $\boldsymbol{\tau}$ is the effective viscous stress tensor of the matrix including both shear and compaction components

$$\boldsymbol{\tau} = \eta\left(\frac{\partial v_{si}}{\partial x_j} + \frac{\partial v_{sj}}{\partial x_i}\right) + \left(\zeta - \frac{2}{3}\eta\right)\delta_{ij}\nabla \cdot \vec{v}_s. \tag{6}$$

$\zeta$ is the volume viscosity. The linearized equation of state for the mixture density is given as

$$\bar{\rho} = \rho_0\left(1 - c_f\varphi\right) \tag{7}$$

with $\rho_0$ as the solid density and $c_f = \frac{\rho_0 - \rho_f}{\rho_0}$. The shear and volume viscosity are given by the equations

$$\eta = \eta_0(1 - \varphi) \tag{8}$$

and

$$\zeta = \eta_0\frac{1-\varphi}{\varphi} \tag{9}$$

where $\eta_0$ is the constant intrinsic shear viscosity of the matrix.

As in both equations (3) and (5) $P_f$ is the fluid pressure (see McKenzie, 1984, Appendix A), these equations can be merged to eliminate the pressure resulting in

$$\vec{v}_f - \vec{v}_s = -\frac{k_0\varphi^{n-1}}{\mu}\left(\rho_0 c_f \vec{g}(1 - \varphi) + \vec{\nabla} \cdot \boldsymbol{\tau}\right). \tag{10}$$

This equation states that the fluid separation flow (i.e. melt segregation velocity) is driven by the buoyancy of the fluid with respect to the solid and the viscous stress in the matrix including compaction and decompaction.

Following Šrámek *et al.* (2010), the Stokes equation (3) can be rewritten by expressing the matrix velocity, $\vec{v}_s$, as the sum of the incompressible flow velocity, $\vec{v}_1$, and the irrotational (compaction) flow

velocity, $\vec{v}_2$, as:

$$\vec{v}_s = \vec{v}_1 + \vec{v}_2 = \begin{pmatrix} \frac{\partial\psi}{\partial z} \\ -\frac{\partial\psi}{\partial x} \end{pmatrix} + \begin{pmatrix} \frac{\partial\chi}{\partial x} \\ \frac{\partial\chi}{\partial z} \end{pmatrix} \tag{11}$$

with $\psi$ as stream function and $\chi$ as the irrotational velocity potential, given as the solution of the Poisson equation

$$\vec{\nabla}^2\chi = \vec{\nabla} \cdot \vec{v}_s. \tag{12}$$

The divergence term $\vec{\nabla} \cdot \vec{v_s}$ can be derived from eqs. 1 and 2 to give

$$\vec{\nabla} \cdot \vec{v}_s = -\vec{\nabla} \cdot \left[\varphi\left(\vec{v}_f - \vec{v}_s\right)\right]. \tag{13}$$

In the small fluid viscosity limit the viscous stresses within the fluid phase are neglected, resulting in a viscous stress tensor in the Stokes equation of the mixture (equ. 5), in which only the stresses in the solid phase are relevant. This is evident from the definition of the viscous stress tensor, which only contains matrix and not fluid viscosities. Melt viscosities of carbonatitic, basaltic or silicic wet or dry melts span a range from $< 1$ Pa s to extreme values up to $10^{14}$ Pa s (see the discussion in Schmeling et al., 2019), while effective viscosities of mafic or silicic partially molten rocks may range between $10^{16}$ Pa s and $10^{20}$ Pa s, depending on melt fraction, stress, and composition. Thus, in most circumstances the small fluid viscosity limit is justified.

In the limit of this small viscosity assumption, inserting the above solid velocity (11) into the viscous stress (6), this into the Stokes equation (5), and taking the curl of the x- and z equations the pressure is eliminated and one gets

$$\left(\frac{\partial^2}{\partial x^2} - \frac{\partial^2}{\partial z^2}\right)\left[\eta_s\left(\frac{\partial^2\psi}{\partial x^2} - \frac{\partial^2\psi}{\partial z^2}\right)\right] + 4\frac{\partial^2}{\partial x\partial z}\left[\eta_s\frac{\partial^2\psi}{\partial x\partial z}\right] = -g\frac{\partial\rho}{\partial x} + A(\chi) \tag{14}$$

with

$$A(\chi) = -2\frac{\partial^2}{\partial x\partial z}\left[\eta_s\left(\frac{\partial^2\chi}{\partial x^2} - \frac{\partial^2\chi}{\partial z^2}\right)\right] + 2\left(\frac{\partial^2}{\partial x^2} - \frac{\partial^2}{\partial z^2}\right)\left[\eta_s\frac{\partial^2\chi}{\partial x\partial z}\right] \tag{14a}$$

To describe the transition from solitary waves to diapirs it is useful to non-dimensionalize the equations. As scaling quantities we use the radius $r$ of the anomaly, the reference viscosity $\eta_0$, and the scaling Stokes sphere velocity (e.g. Turcotte & Schubert, 1982) based on the maximum porosity of the anomaly $\varphi_{max}$

$$v_{St} = C_{St}\frac{\varphi_{max}\Delta\rho g r^2}{\eta_0} \tag{15}$$

resulting to the following non-dimensionalization where non-dimensional quantities are primed:

$$(x, z) = (x', z') \cdot r, \qquad \vec{v}_{s,f} = \vec{v}_{s,f}' \cdot v_{St}, \qquad t = t' \cdot \frac{r}{v_{St}}, \qquad (\tau_{ij}, P) = (\boldsymbol{\tau}', P') \cdot \frac{\eta_0 v_{St}}{r},$$

$$(\eta, \zeta) = (\eta', \zeta') \cdot \eta_0, \qquad (\psi, \chi) = (\psi', \chi') \cdot r v_{St} \tag{16}$$

For $r$ the half width of the prescribed initial perturbation, consisting of a 2D Gaussian bell, is chosen. This is reasonable as the rising velocity in our code is best described by the Stokes velocity, using this radius. The exact shape of the perturbation is given later in the model setup.

$C_{St}$ is calculated by using the analytic solution of an infinite Stokes cylinder within another cylinder (Popov and Sobolev (2008), based on the drag force derived by Slezkin (1955)), because, due to boundary effects, the cylinder gets effectively slowed. $C_{St}$ is calculated using $C_{St} = \ln(k) - \frac{k^2-1}{k^2+1}$, where k is the ratio of outer cylinder's to inner cylinder's radius. For our model setup $C_{St}$ is equal to 0.17.

With these rules the Darcy equation (10) is given in non-dimensional form

$$\vec{v}_f{}' - \vec{v}_s{}' = -\frac{\delta_c^2}{r^2}\frac{1}{\tilde{\eta}'\varphi}\left(\vec{e}_z\frac{(1-\varphi)}{\varphi_{max}} + \overrightarrow{\nabla'}\cdot\boldsymbol{\tau}'\right) \tag{17}$$

where $\vec{e}_z$ is the unit vector in z-direction and $\tilde{\eta}'$ is equal to $\zeta' + \frac{4}{3}\eta'$. The momentum equation of the mixture (12) is given by

$$\left(\frac{\partial^2}{\partial x'^2} - \frac{\partial^2}{\partial z'^2}\right)\left[\eta_s{}'\left(\frac{\partial^2\psi'}{\partial x'^2} - \frac{\partial^2\psi'}{\partial z'^2}\right)\right] + 4\frac{\partial^2}{\partial x'\partial z'}\left[\eta_s{}'\frac{\partial^2\psi'}{\partial x'\partial z'}\right] = \frac{1}{\varphi_{max}}\frac{\partial\varphi}{\partial x'} + A'(\chi'). \tag{18}$$

$\delta_c^2/r^2$ in equation (17) is the squared ratio of compaction length $\delta_c$ to the system length scale $r$, which is the main parameter describing our system. The compaction length is a natural length scale emerging from the problem and of particular importance in our context, because 2D porosity waves have half width radii of the order of $3\cdot\delta_c$ to $5\cdot\delta_c$ (Simpson and Spiegelman, 2011). It is defined as:

$$\delta_c = \sqrt{\frac{\zeta + \frac{4}{3}\eta}{\mu}k_\varphi} \tag{19}$$

All quantities in the other equations are simply replaced by their non-dimensional primed equivalents (eqs. (1), (2), (6), (11), (12), (13), and (14a)).

We now compare the two limits, where segregation or two-phase flow dominates (solitary wave regime), and where fluid and solid rise together with the same velocity as partially molten bodies, which we identify with the diapir regime. We compare the characteristic segregation velocity within solitary waves, which scales as

$$v_{sgr} \approx \frac{k_0\varphi_{max}{}^{n-1}}{\mu}\left(\Delta\rho g(1-\varphi_{max}) - \overrightarrow{\nabla'}\cdot\boldsymbol{\tau}\right) = C_{sgr}\frac{k_0\varphi_{max}{}^{n-1}\Delta\rho g(1-\varphi_{max})}{\mu} \tag{20}$$

where $C_{sgr}$ is of the order ½ for 2D solitary waves (Schmeling, 2000), with the characteristic Stokes sphere rising velocity given by (15). The ratio of these is given by

$$\frac{v_{sgr}}{v_{st}} = \frac{C_{sgr}}{C_{st}}\frac{\delta_{c0}^2}{r^2}\frac{\varphi_{max}{}^{n-2}(1-\varphi_{max})}{\widetilde{\eta_0}'\varphi_0^n} \tag{21}$$

Here $\widetilde{\eta_0}'$ refers to $\tilde{\eta}'$ for the background porosity $\varphi_0$ and $\delta_{c0}$ to the compaction length of the background porosity. In contrast to Scott (1988), who varies the volume viscosity in his model series, we vary the ratio of initial Stokes radius to compaction length.

Thus, in the solitary wave limit

$$\frac{C_{sgr}}{C_{st}}\frac{\delta_{c0}^2}{r^2}\frac{\varphi_{max}{}^{n-2}(1-\varphi_{max})}{\widetilde{\eta_0}'\varphi_0^n} \gg 1 \tag{22}$$

Darcy's law (17) results in large segregation velocity, which scales as

$$v_{sgr}' = \frac{C_{sgr}}{C_{st}} \frac{\delta_{c0}^2}{r^2} \frac{\varphi_{max}^{n-2}(1-\varphi_{max})}{\widetilde{\eta_0}' \varphi_0^n} \tag{23}$$

From equation (13) it follows that the irrotational part of the matrix velocity scales with

$$v_1 \approx -\varphi_{max} v_{sgr} \tag{24}$$

while the rotational part is given by (18): In that equation $A'$ scales with $\chi'$, which, via equation (12) and (13), scale with $v_{sgr}$, i.e. with $\delta_{c0}^2/r^2$. In other words, the second term on the RHS of (18) dominates for small $r^2/\delta_{c0}^2$ as the first term is of the order 1. Thus, the rotational matrix velocity has the same order as the irrotational compaction velocity and serves to accommodate the compaction flow. In this limit the buoyancy term in equation (18), $\frac{1}{\varphi_{max}} \frac{\partial \varphi}{\partial x'}$, is of vanishing importance for the matrix velocity and the matrix velocity, $\vec{v}_1 + \vec{v}_2$, is of the order of $\varphi_{max} v_{sgr}$. In the small porosity limit, matrix velocities are negligible with respect to fluid velocities.

In the diapir limit,

$$\frac{C_{sgr}}{C_{st}} \frac{r^2}{\delta_c^2} \frac{\varphi_{max}^{n-2}(1-\varphi_{max})}{\widetilde{\eta_0}' \varphi_0^n} \ll 1 \tag{25}$$

and equation (17) predict vanishing segregation velocities. As $A'$ and $\chi'$ scale with $r^2/\delta_{c0}^2$ , both vanish in the diapir limit, no irrotational matrix velocity occurs and equ. (18) reduces to the classical biharmonic equation (i.e. Stokes equation) driven by melt buoyancy and describing classical diapiric ascent. Segregation velocities are negligible with respect to matrix velocities.

In Fig. 1 the results of this simple analysis are shown, where we calculated the velocity ratios as a function of initial perturbation radius for several perturbation radii. In our models we use a $\varphi_{max}$ of 2%, for which we get a switch from solitary wave to diapir dominant behavior at $r = 48 \cdot \delta_c$. Smaller amplitudes lead to a switch at a smaller radius and larger amplitudes to a switch at a larger radius.

## 2.2 Model setup

The model consists of a $L' \times L'$ box with a background porosity, $\varphi_0$, of 0.5%. $L'$ is the non-dimensional side length of the box and equal to 6 times the initial radius of the perturbation. As initial condition a non-dimensional Gaussian bell-shaped porosity anomaly is placed in the middle of the model at $x_0' = 3$ and $z_0' = 3$. The Gaussian wave is given by

$$\varphi = \varphi_{max} \cdot \exp\left(-\left(\frac{x'-x_0'}{w'}\right)^2 - \left(\frac{z'-z_0'}{w'}\right)^2\right) \tag{26}$$

Where $\varphi_{max}$ is the amplitude equal to 0.02 in our models and $w'$ corresponds to the width where $\varphi$ has reached $\varphi_{max}/e$. In our case $w'$ is equal to 1.2.

In our model series we vary the ratio of Stokes radius to compaction length from 1.8 to 48 to explore the transition from solitary wave towards diapiric regime. The resolution of the models is chosen to be at least $201 \times 201$ grid points and was increased for higher ratios of Stokes radius to compaction length so that the compaction length is resolved by at least 3-4 grid points.

At the top and the bottom domain boundaries, we prescribe an out- and inflow for both melt and solid,
respectively, to prevent melt accumulations at the top. The segregation velocity of the background porosity $\varphi_0$ is calculated using equation (17) without the viscous stress term. The corresponding matrix velocity is calculated using the conservation of mass.

At the sides we enforce no horizontal flux boundary conditions. The permeability-porosity relation exponent in our models is always $n = 3$.

To run models for a longer, practically infinite, amount of time we let the models coordinate system follow the maximum melt fraction.

### 2.3 Numerical approach

We discretize the set of equations using finite differences on a staggered grid and solve the system using the code FDCON (Schmeling et al., 2019). Starting from the prescribed initial condition for $\varphi$, and
assuming $A'(\chi') = 0$ at time 0, the time loop is entered and the biharmonic equation (19) is solved for $\psi'$ by Cholesky decomposition, from which $\vec{v}_1'$ is derived. Together with $\vec{v}_2'$ the resulting solid velocity is used to determine the viscous stress term in the segregation velocity equation (17). This equation and the melt mass equation (1) are solved iteratively with strong underrelaxation for $\varphi$ and $\vec{v}_f' - \vec{v}_s'$ for the new time step using upwind and an implicit formulation of equ. (1). During this internal iteration these
quantities are used, via equ. (13), to give $\vec{\nabla} \cdot \vec{v}_s$, the divergence of the matrix velocity, which is needed in the viscous stress term (equ. 6). After convergence $\vec{\nabla} \cdot \vec{v}_s$ is used via equ. (12) to determine $\chi$ by LU-decomposition and then to get $\vec{v}_2'$. Now $A'(\chi')$ can be determined to be used on the RHS of equ (18). The procedure is then repeated upon entering the next time step.

Time steps are dynamically adjusted by the Courant criterion times 0.2 based on the fastest velocity,
either melt or solid.

The model resolution is a critical parameter in this kind of numerical calculations and should always be kept in mind. With increasing length scale ratio, the compaction length in the model gets smaller and the resolution needs to be increased to keep it equally resolved.

According to several authors (e.g. Räss et al., 2019; Keller et al., 2013), the compaction length should
be at least resolved by 4-8 grid points to solve for waves sufficiently accurately. For small length scale ratios this is no problem, where, with a model resolution of $201 \times 201$, up to nearly 30 grid points per compaction length can be achieved. The highest resolution our code can run is $601 \times 601$, which is

enough to resolve the compaction length by three grid points for the model with a length scale ratio of 48. Everything above that cannot be sufficiently resolved with respect to studying solitary waves.

Fig. 2 shows the resulting models for a length scale ratio of 12 for six different resolutions. The model states after $\varphi_{max}$ has risen approximately 0.25 times the initial Stokes radius ($t' = 0.25$) are shown. With increasing resolution, the maximum melt fraction increases strongly from $101 \times 101$ to $401 \times 401$ by approximately 20% but the velocity of $\varphi_{max}$ decreases by 7% (not shown in the figure). Both values converge for resolutions higher than 51x51, corresponding to $\delta_c/dx = 1$. Even though the compaction length is not sufficiently resolved in Fig. 2d, one can still observe the main features of the model: A main solitary wave has emerged from the original gaussian perturbation and secondary porosity waves are beginning to emerge within its wake. Even with $\delta_c/dx = 1$ these features can be observed but are clearly underresolved. With even lower resolutions accumulations at the top of the perturbation can be seen, which can be broadly interpreted as the attempt of a solitary wave to build up. With $\delta_c/dx = 0.24$, the model is too coarse and the results cannot be trusted anymore.

The solitary waves modeled with our code have been compared to the semi-analytical solution of Simpson & Spiegelman (2011), and more benchmarking was carried out in Dohmen et al. (2019).

In a single-phase flow case, where the melt is not allowed to move relatively to the solid, the initial perturbation ascends, shortly after beginning, with a velocity of 0.95 times the calculated Stokes velocity, and then slowly decreases as the original Gauss-shaped wave deforms and loses in amplitude.

## 3 Results

### 3.1 The transition from porosity wave to diapirism: Varying the initial wave radius

In this model series we vary the initial wave radius to cover the transition from porosity waves towards diapirism. As a reminder, due to our scaling the initial wave has always the same size w.r.t. the model box, and "increasing the initial wave radius" is equivalent to decreasing the compaction length or the size of the emerging solitary waves w.r.t. the model box. In Fig. 3 the models are shown at $t' = 0.2$. For small radii ($r \leq 12 \cdot \delta_c$) a single porosity wave emerges from the original perturbation. The melt that is not situated within the emerging wave is left behind and has, for the most part, already left the model region. For $r = 2.4 \cdot \delta_c$ the emerged solitary wave is about the size of the initial perturbation and even smaller radii would lead to too big waves that would not fit into the model. With increasing radius, the emerging solitary wave gets smaller. With $r = 12 \cdot \delta_c$, the resulting wave has just a size of ~20% the initial perturbation size.

We compare the observed solitary wave velocities of Fig. 3b-e to equivalent Stokes velocities for a diapir based on equation (15). While the dimensional Stokes velocity of a porosity anomaly is proportional to the amplitude of porosity and the square of the radius, the non-dimensional Stokes velocity is always equal to 1. In Fig. 4 this non-dimensional Stokes velocity is indicated by the dashed

line with the value 1. The colored lines give 2D solitary wave velocities with their appropriate radii, given by Simpson & Spiegelman (2011), normalized by the Stokes velocity corresponding to different initial perturbation radii. These semi analytical solutions are in good agreement to our solitary wave models and differ only by 3-5% percent in velocity, as already shown in Dohmen et al. (2019). The velocities in this figure correspond to ratios of solitary wave velocity to initial perturbation Stokes velocity. Inspection of Fig. 4 reveals that for the first four cases of Fig. 3b-e with radii smaller or equal $12 \cdot \delta_c$ the phase velocities are always larger than the Stokes velocity. For example, for $r = 12 \cdot \delta_c$, an emerging solitary wave with a typical radius of $4.5 \cdot \delta_c$ has a higher phase velocity than a $r = 12 \cdot \delta_c$ melt anomaly rising by Stokes flow. Thus, the cases are always in the solitary wave regime.

For greater radii (e.g. $r = 18 \cdot \delta_c - 30 \cdot \delta_c$, Fig. 3e-g) the phase velocities of solitary waves are of the order of the Stokes velocity (see Fig. 4) and they therefore need more time to separate from the remaining melt of the initial perturbation, still rising with order of Stokes velocity. The amount of melt accommodated within the main solitary wave is just a small percentage of the original perturbation and secondary waves evolve in its remains. With further ascending, more and more solitary waves build up and the former perturbation will sooner or later consist of solitary waves in an ordered cluster or a formation. This formation elongates during ascent as the main wave has a larger amplitude than all the following waves, whose amplitudes are also decreasing with depth, as a higher proportion of melt accumulated at the top of the perturbation. Similar formations of strongly elongated fingers can be also observed in 3D as shown by Räss et al. (2019) who used decompaction weakening. In the models with smaller radii, the main solitary wave consisted of the majority of melt originally situated within the perturbation and the emergence of secondary waves turns out zero or small, but with greater radii enough melt is left behind to observe the emergence of second and higher generations of solitary waves.

For greater radii (e.g. $r = 24 \cdot \delta_c - 48 \cdot \delta_c$, Fig. 3 f – j) the phase velocities of solitary waves are almost equal to the Stokes velocity (See Fig. 4). This leads to almost no separation after $t' = 0.2$. While for $r = 36 \cdot \delta_c$ a solitary wave has already built up and is rising just ahead of the perturbation, for $r = 42 \cdot \delta_c$ and $r = 48 \cdot \delta_c$ just the accumulation of melt at the top of the perturbation can be observed, which will eventually lead to a solitary wave. Secondary waves also build up with higher runtimes, as can be already seen for $r = 36 \cdot \delta_c$.

For even greater radii the compaction length cannot be sufficiently resolved with our approach, but tests with not sufficiently resolved models have shown that solitary waves can be observed for $r \geq 48 \cdot \delta_c$. At some point they do no longer appear, probably due to lack of sufficient resolution, but our tests show that solitary waves should always emerge, even if its phase velocity is way below the Stokes velocity. As long as the ascending time is long enough and melt is able to move separately to the matrix, independently of segregation velocity, a diapir will evolve into a swarm of a certain number of solitary waves, based on the compaction length. Because the phase velocities of each small solitary wave is

small compared to the Stokes velocity of the full swarm we consider such a rising formation of melt as a large scale diapir.

Fig. 3l shows the required time for the initial perturbation to build up a solitary wave. This status is achieved after the dispersion relation of the main wave reaches a point from where it follows the solitary wave dispersion relation. This time increases nearly linearly for small radii ($r \leq 48 \cdot \delta_c$) but increases non-linearly for greater radii. This might be due to lack of proper resolution, but a non-linear trend can be already observed for small radii. The transition time for radii smaller than $30 \cdot \delta_c$ is smaller than 0.2, the time at which the models in Fig. 3b-j are shown. The other models already show solitary wave like blobs but did not yet reach their final form.

A classical diapir will evolve only in cases with zero compaction length ($r = \infty \cdot \delta_c$), i.e., melt is not able to move w.r.t. the matrix (Fig. 3k). Here, no focusing into solitary waves can be observed and transition time is infinity.

Summarizing Fig. 4, the comparison of Stokes and porosity wave velocities explains well our observations shown in Fig. 3: For small initial radii the solitary wave velocity is clearly higher and will therefore build up and separate from the melt left behind quickly. For cases with approximately equal perturbation to solitary wave radius only one solitary wave will build up, which includes most of the melt of the initial perturbation. With increasing perturbation radius, the velocity ratio decreases and multiple solitary waves, requiring more time, will emerge, each including only a fraction of the melt originally situated in the initial perturbation. But even with velocity ratios smaller than 1, solitary waves emerge and, not able to separate, rise just ahead of the remains, slowly elongating the initial perturbation.

**3.2 Effects on the mass flux**

It is important to study the partitioning between rising melt and solid mass fluxes in partially molten magmatic systems because melts and solids are carriers of different chemical components. Within our Boussinesq approximation we may neglect the density differences between solid and melt. Then our models allow to evaluate vertical mass fluxes of solid or fluid by quantifying the vertical velocity components multiplied with the melt or solid fractions, respectively:

$$q'_{sz} = (1 - \varphi) \cdot v'_{sz}$$
$$q'_{fz} = \varphi \cdot v'_{fz}.$$

(29)

Horizontal profiles of the mass fluxes through rising melt bodies at the vertical positions of maximum melt fraction at timesteps where the main wave has just reached the status of a solitary wave are calculated (Fig. 5).

The mass fluxes of solid and fluid are strongly affected by the change of the initial radius from the solitary wave regime towards the diapiric regime. For $r = 2.4 \cdot \delta_c$, where we observe a solitary wave, the fluid has its peak mass flux in the middle of the wave and the solid is going downwards, against the

phase velocity. In the center the fluid flux is about 10 times higher than the solid net flux. The upward
flow in the center is balanced by the matrix dominated downward flow inside and outside the wave. For
$r = 12 \cdot \delta_c$ the wave area is much smaller and the ratio between solid and fluid flux is still around the
order of 10. At the boundary of the wave the solid is nearly not moving at all, but a minimum can be
observed within the center of it. For $r' = 24 \cdot \delta_c$ the solid flux is just above zero in the center and
increases to a maximum towards the flanks of the wave, that is still about ten times smaller than the
maximum fluid flux.

With $r' = 48 \cdot \delta_c$ the solid flux is just about three times smaller than the fluid flux, but most of the
material ascent is accomplished by the solid. This suggests that diapiric rise begins to dominate.

The transition from solitary waves towards diapirism on qualitative model observations was so far only
based on observations. We now invoke a more quantitative criterion. In a horizontal line passing through
the anomaly's porosity maximum we define the total vertical mass flux of the rising magma body by
$\int_{\varphi > \varphi_0} (q_f + q_s) dx$ where the integration is carried out only in the region of increased porosity $\varphi > \varphi_0$.
This mass flux is partitioned between the fluid mass flux, $\int_{\varphi > \varphi_0} q_f dx$, and the solid mass flux,
$\int_{\varphi > \varphi_0} q_s dx$. With these we define the partition coefficients

$$C_{\text{soli}} = \frac{\int_{\varphi > \varphi_0} q_f dx}{\int_{\varphi > \varphi_0} (q_f + q_s) dx} \tag{30}$$

and

$$C_{\text{dia}} = \frac{\int_{\varphi > \varphi_0} q_s dx}{\int_{\varphi > \varphi_0} (q_f + q_s) dx} \tag{31}$$

The sum $C_{\text{soli}} + C_{\text{dia}}$ is always 1 and if $C_{\text{soli}} > C_{\text{dia}}$ then the solitary wave proportion is dominant, while
for $C_{\text{soli}} < C_{\text{dia}}$ diapirism is dominant. In Fig. 6a these partition coefficients for several initial radii are
shown. In red are the diapir and in blue the solitary wave partition coefficients.

For $r = 1.8 \cdot \delta_c$, $C_{\text{soli}}$ is equal to 5 and $C_{\text{dia}}$ is equal to -4, i.e. we have a downward solid flux. With
increasing radius $C_{\text{dia}}$ increases until it changes its sign, and the matrix flows upward, at $r \approx 20 \cdot \delta_c$. It
eventually becomes bigger than $C_{\text{soli}}$ at $r = 36 \cdot \delta_c$ and then approaches 1 for bigger radii. $C_{\text{soli}}$
changes so that the sum of both is always equal to 1. Even though diapirism is dominant for $r > 36 \cdot \delta_c$,
we still observe solitary waves, yet their phase velocities are much smaller than the large-scale rising
velocities of the full melt formation.

The ratio of maximum fluid velocity (i.e. $\overrightarrow{v_f}$) to absolute matrix velocity (Fig. 6b) shows, that for small
radii, where $C_{\text{soli}} \gg C_{\text{dia}}$, this ratio is approximately constant with a high value of about 100. The
absolute velocity maxima itself are not constant but decrease with the same rate until the switch of

negative to positive matrix mass flux, where the absolute matrix velocity starts to increase, while the fluid velocity keeps decreasing. At this zero crossing we would expect a ratio of infinity, but while the zero crossing takes place within the center of the solitary wave, other regions near the wave still have finite vertical velocities. This switch from negative to positive mass flux was already observed by Scott (1988), but while they changed the viscosity ratio as an independent constant model parameter, we change the radius and keep the viscosity law the same, still evolving with $\varphi$. Both describe the transition from a two-phase limit towards the Stokes limit, but in our formulation, we are able to reach the Stokes limit while Scott's formulation (1988) is restricted to two-phase flow. With even greater radii the velocity ratio will eventually converge towards 1, where melt is no longer able to move relatively to the matrix (i.e. $\vec{v_f} = \vec{v_s}$) and material will be transported collectively as in single-phase flow. These last models are not sufficiently resolved to obtain leading and secondary solitary waves, but still show the expected behavior in terms of macroscopically rising partially molten diapir.

Based on these observations, the evolution of these models can be divided into two regimes: (1) In the solitary wave regime ($r \leq 36 \cdot \delta_c$) $C_{\mathrm{soli}}$ is larger than $C_{\mathrm{dia}}$ and the initial perturbation emerges into waves that have the properties of solitary waves and ascend with constant velocity and staying in shape. This regime can be further divided into 1a ($r < 20 \cdot \delta_c$), where the solid mass flux is negative, and 1b ($20 \cdot \delta_c \leq r < 36 \cdot \delta_c$), where the solid moves upwards with the melt. Waves in these regimes are very similar but the further we are in regime 1a the less solitary waves will emerge out of the initial perturbation. For radii smaller than about $4.8 \cdot \delta_c$ only one wave will merge. In regime 1b the perturbation will always emerge into multiple solitary waves.

In the diapirism-dominated regime (2) ($r \geq 36 \cdot \delta_c$), $C_{\mathrm{dia}}$ is larger than $C_{\mathrm{soli}}$ but, as the fluid melt is still able to move relatively to the solid matrix, solitary waves build up and the whole partially molten region will evolve into a swarm of them. The phase velocities of these waves are very small compared to the Stokes velocity of the perturbation and the whole swarm will rise as a diapir, whose buoyancy is still comparable to the buoyancy of the initial perturbation's.

The endmember of the second regime can be reached by prohibiting the relative movement of fluid ($r = \infty \cdot \delta_c$), for which the compaction length has not to be sufficiently resolved. In this regime the initial perturbation will not disintegrate into solitary waves but rise as a well-formed partially molten diapir. In every other case, in the present model, where fluid is able to move w.r.t. the solid, at some point all diapirs will evolve into a swarm of solitary waves which can be infinitely small compared to the initial perturbation. However, this is expected to happen only after a long distance of diapiric rise. In cases where the size of solitary waves is comparable to the perturbation (e.g. regime (1)) this will occur sooner and in cases, where solitary waves are much smaller, later. Their observation is mostly limited by resolution. For models that allow for the diapir to grow (e.g. Keller et al., 2013) they may not dissolve into solitary waves, as it approaches the single-phase limit.

## 4 Discussion

### 4.1. Application to nature

While in our models the perturbation size in terms of compaction lengths was systematically varied but kept constant within in each model, our results might also be applicable to natural cases in which the compaction length varies vertically. In the case of compaction length decreasing with ascent a porosity anomaly might start rising as a solitary wave but then at some point might enter the second regime where diapiric rise is dominant. If this boundary is sharp, the solitary wave might disintegrate into several smaller solitary waves that rise as a diapiric swarm. If the boundary is a continuous transition the wave should slowly shrink and become slower. The melt left behind might also evolve into secondary solitary waves.

A decreasing compaction length could be accomplished by decreasing the matrix viscosity or the permeability, or by increasing the fluid viscosity. Decreasing matrix viscosity might be for example explainable by local heterogeneities, temperature anomalies for example due to secondary convective overturns in the asthenosphere or by a vertical gradient of water content, which may be the result of melt segregation aided volatile enrichment at shallow depths in magmatic systems. This could lead to the propagation of magma-filled cracks (Rubin, 1995) as already pointed out in Connolly & Podladchikov (1998). The latter authors have looked at the effects of rheology on compaction-driven fluid flow and came to similar results for an upward weakening scenario. The decrease of permeability due to decrease in background porosity might be an alternative explanation. In the hypothetic case of a porosity wave reaching the top of partially molten region within the Earth's upper mantle or lower crust, the background porosity might decrease which would most certainly lead to focusing, because the compaction length will decrease, and eventually, when reaching melt free rocks, the solitary waves might be small enough and its amplitude might be high enough to trigger the initiation of dykes.

Even though most diapirs should, according to our models, disintegrate into numerous solitary waves, not all will inevitably. Within regime (1) solitary waves are possible and most probably expected but the deeper we are in regime (2) the less expected is the disintegration because a long time is needed to build up. In nature, different from our models, they cannot rise for an infinite amount of time. The time needed to build up a solitary wave increases non-linearly with $r$ (c.f. Fig. 3 l). For example, while for $r = 4.8 \cdot \delta_c$ a solitary wave is completely evolved after $t' = 0.02$, for $r = 48 \cdot \delta_c$ it needs until $t' = 0.4$, i.e., equivalent to the diapiric rise time necessary to ascend the distance approximately half the initial radii. Additionally, as already pointed out, if a model setup allows for the diapir to grow, it could approach the single-phase flow, prohibiting the emergence of solitary waves (cf. Keller et al., 2013).

### 4.2. Model limitations

The introduced partition coefficients help to distinguish whether solitary wave or diapiric rise is dominant but cannot be solely consulted whether a solitary wave or a diapir can be expected. As the

fluid velocity and flux is still very high in the waves center for diapiric dominant cases, small solitary waves will build up. However, the net mass flux is dominated by the large scale rising solid, and the formation time of small solitary waves might be long. Additionally, the internal circulation of diapirs can be faster than the phase velocity which would smear out the emergence of solitary waves and not allow for them to emerge. Due to limitations of our model, we are not able to reach regions where solitary waves are small enough and their phase velocity slow enough to observe this.

While the minimum size of solitary waves in nature might be in some way limited by the grain size, in numerical models the minimum size is limited by the model's resolution. We restrict our models in this study to cases where the compaction length is at least resolved by 3 grid lengths $dx$ (i.e. $\delta_c \geq 3 \cdot dx$) to get fairly resolved solitary waves, but they can be also observed for much worse resolved compaction lengths. The resolution test (Fig. 2) shows that, even though they are not solved decently, probable solitary waves can be observed for cases with $\delta_c = dx$. Smaller resolutions can show indications of solitary waves but should not be trusted as other tests (not shown here) with similar resolutions result in spurious channeling. For very poorly resolved compaction lengths ($\delta_c < 0.25 \cdot dx$ for our models) no indications of solitary waves can be observed, and the partially molten perturbation ascends as a diapir. The deeper we are in regime 2, the more dominant are the dynamics of diapirism on a length scale of r compared to Darcy flow or solitary waves on the unresolved length scale of $\delta_c$. Thus, two-phase flow, either Darcy flow or solitary waves, becomes negligible for $r \gg \delta_c$ and partially molten diapirs can be regarded as well resolved.

**5 Conclusion**

This work shows, that depending on the extent of a partially molten region within the Earth, the resulting ascent of melt may not only occur by solitary waves or by diapirs, but by a composed mechanism, where a diapir splits up into numerous solitary waves. Their phase velocities might become so slow that the whole swarm will ascend as a diapir, just slowly elongating due to the main solitary wave having a higher amplitude and therefore higher phase velocity than the following ones. Depending on the ratio of the melt anomalies size to the compaction length, or rather the models length scale to compaction length ratio, we can classify the ascent behavior into two different regimes using mass flux and velocity of matrix and melt: (1a + b) Solitary wave a and b, and (2) diapirism-dominated. In regime 1a the matrix sinks with respect to the rising melt, in 1b also the matrix rises, but very slowly. The further we are in this regime the less solitary waves will emerge out of the initial perturbation until, eventually, only one solitary wave will emerge. On first order these regimes can be explained by comparing Stokes velocity of the rising perturbation with the solitary waves phase velocity. If the solitary wave velocity is higher than the Stokes velocity a solitary wave will evolve and, if lower, diapirism is dominant, but still solitary waves will build up if the ascending time is long enough. The deeper we are in regime 2, the more time is needed to build up solitary waves and the less likely it is that they will appear in nature. The

endmember of regime (2), pure diapirism, can be reached if fluid is not allowed to move separately to the matrix.

Especially around the transition of the regimes numerical resolution plays an important role as the compaction length may be under-resolved to allow for the emergence of solitary waves. Hence it should be generally important for two-phase flow models to inspect whether solitary waves are expected and if so, do they have a major influence on the conclusions made.

**Code availability**

The used finite difference code, FDCON, is available on request.

**Author Contribution**

Janik Dohmen wrote this article and carried out all models shown here. Harro Schmeling helped preparing this article and initialized this project.

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

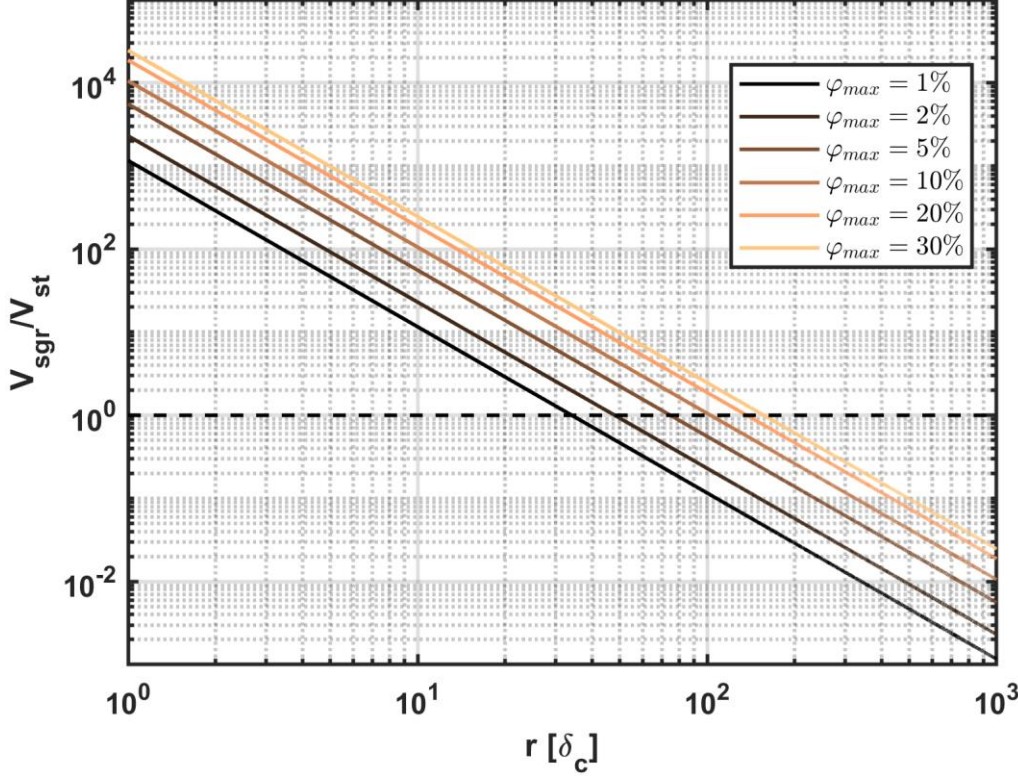

**Fig. 1: The segregation to Stokes velocity ratio, following equation (21), is given as a function of initial perturbation radius $r$ in terms of compaction length $\delta_c$. Each colored line refers to different values of perturbation amplitude $\varphi_{max}$, given in the legend.**

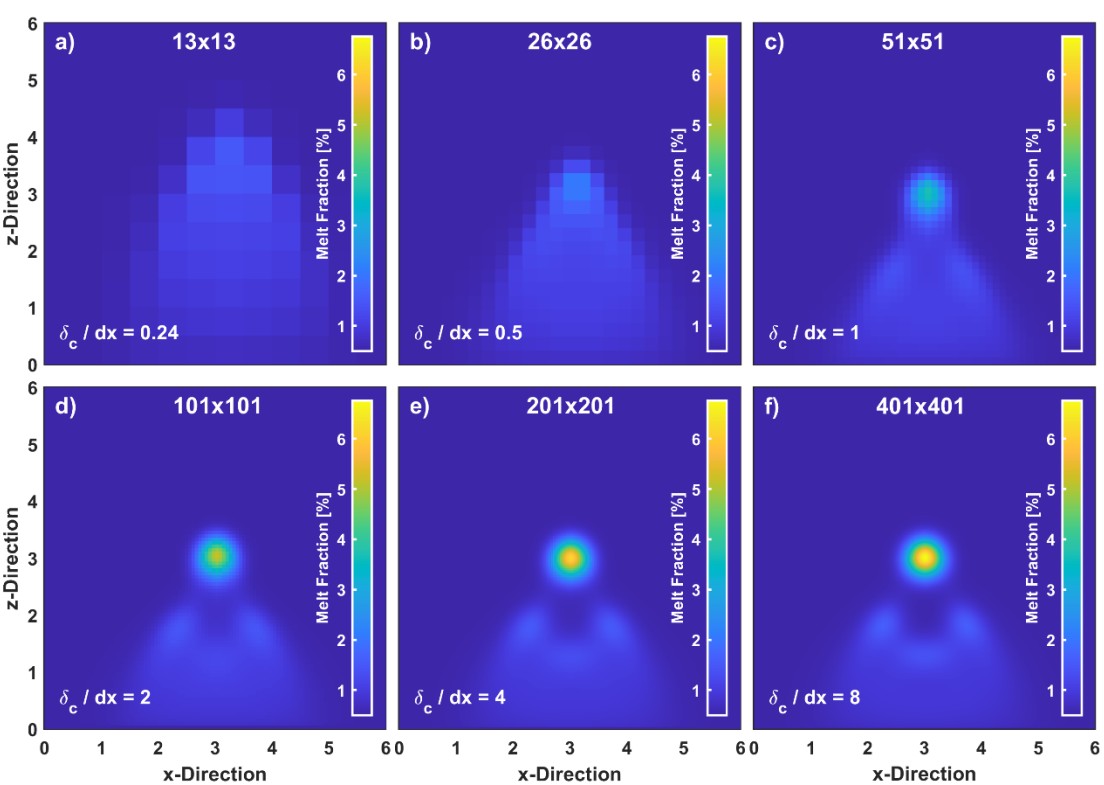

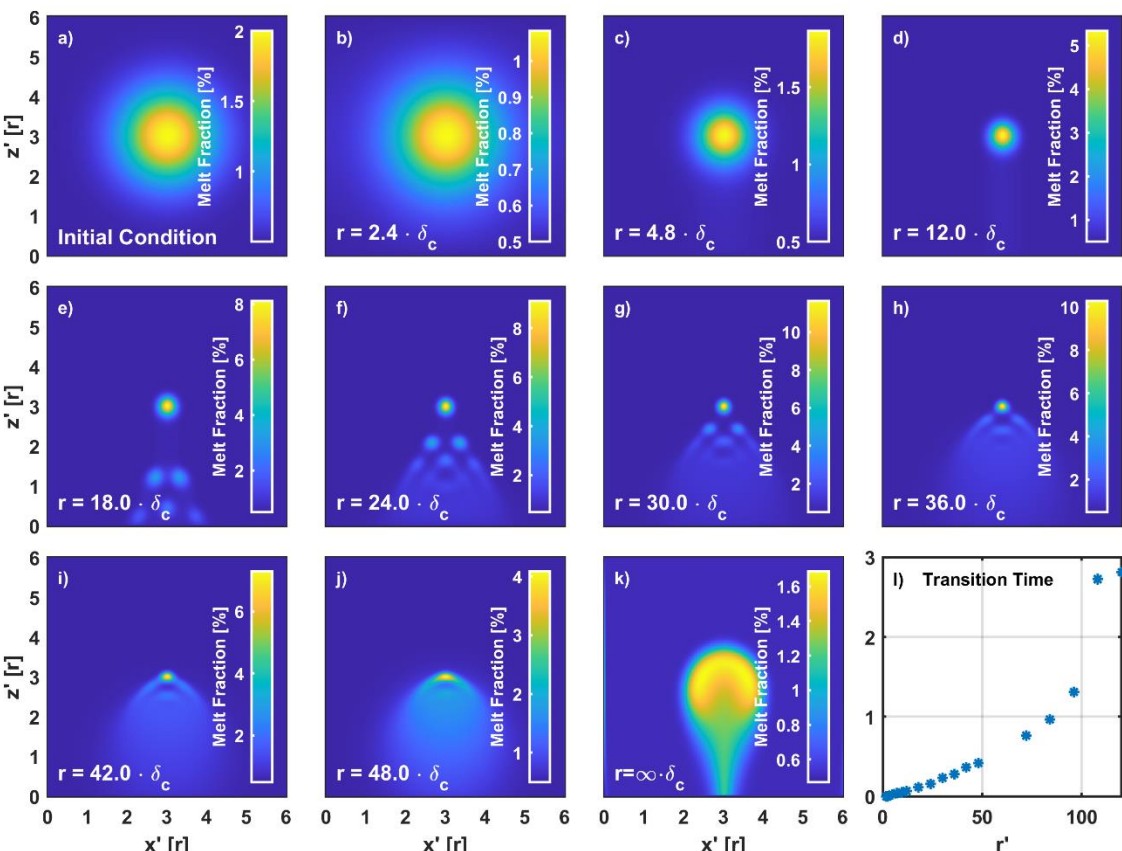

**Fig. 2: The six panels depict a model with an initial perturbation radius of 12 times the compaction length but with different numerical grid resolutions: a) 13x13 b) 26x26 c) 51x51 d) 101x101, e) 201x201, f) 401x401. In the lower left corner in each figure the size of the compaction length in terms of grid length is given.**

**Fig. 3: Melt ascent morphology as function of initial perturbation radius in terms of compaction length. a) Initial conditions of the model valid for all cases apart of the change in compaction**

**length. b-j) Melt fraction distribution after $t' = 0.2$ for length scale ratios varying between 2.4 and 48. k) Diapiric rise resulting from a compaction length of zero at $t' = 9$. l) Models' transition time as function of length scale ratios varying between 1.8 and 120. The transition time gives the time after which the main wave has reached a solitary wave status.**

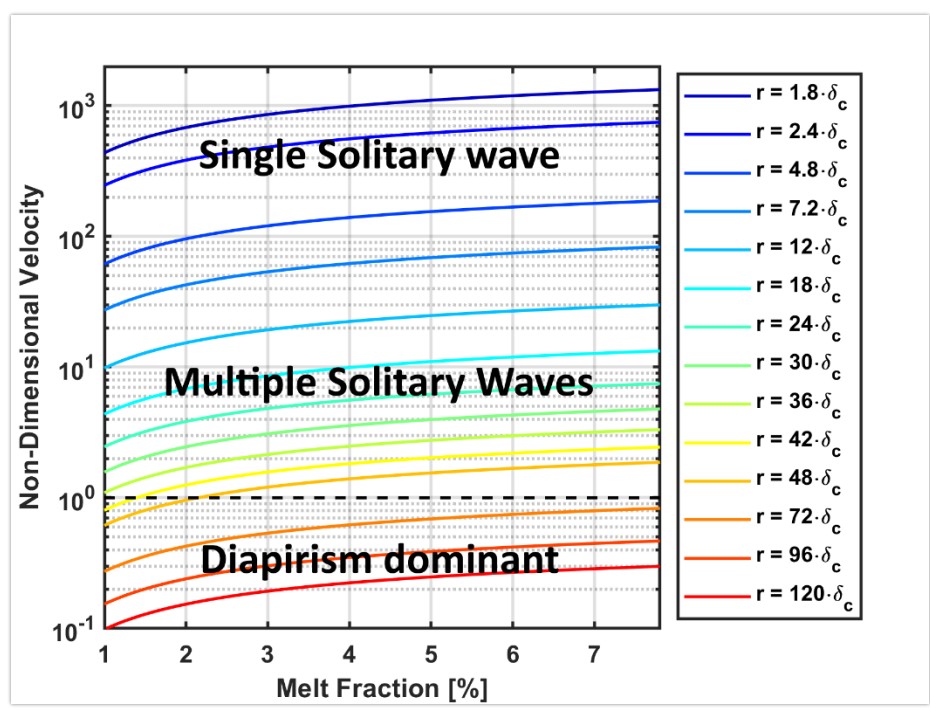

Fig. 4: The dashed line marks the velocity of the Stokes sphere ($v' = 1$). The colored lines refer to the velocity of a 2D solitary wave, calculated semi-analytically by Simpson & Spiegelman (2011), in our non-dimensionalization, based on the radii shown in the legend.

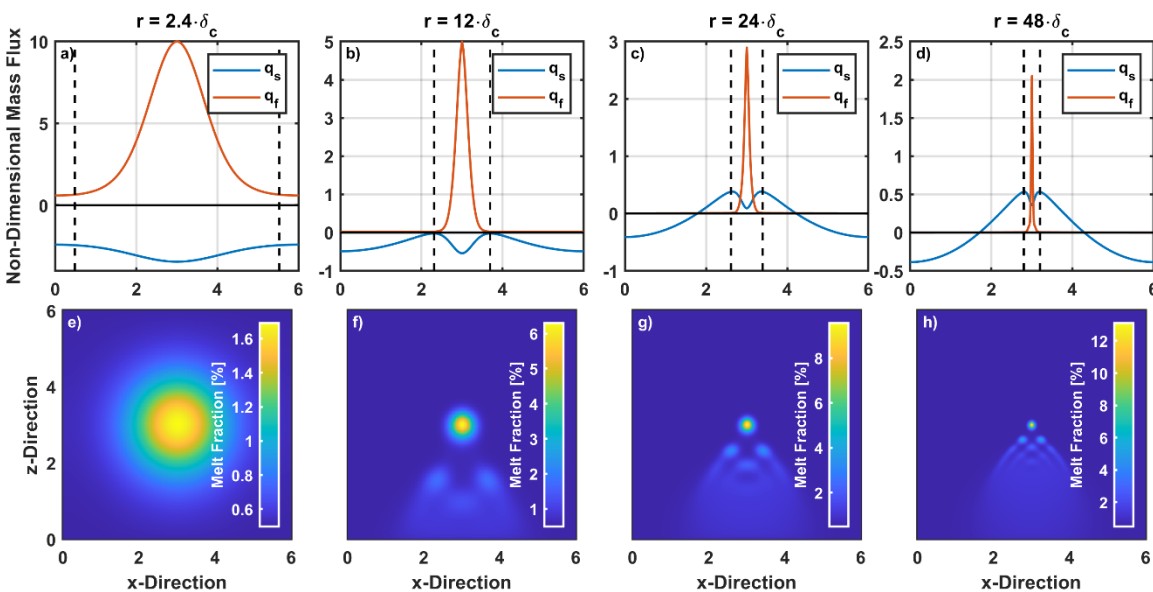

Fig. 5: The upper row panels depict the solid and fluid mass fluxes of a horizontal line cutting through the maximum melt fraction at timesteps where the main wave has just reached the status of a solitary wave. These timesteps are $t' = 0.02; 0.068; 0.155; 0.416$ from left to right, respectively. The bottom row panels depict the corresponding melt porosity fields. All quantities shown are non-dimensional.

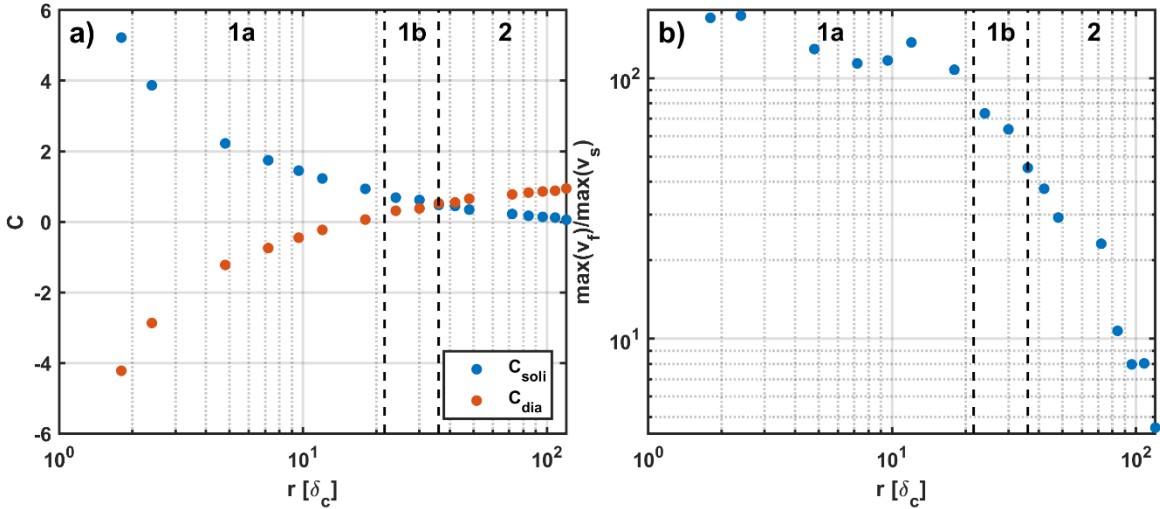

**Quantitative parameters as function of initial perturbation radius in terms of compaction length.**
**a) Solitary wave (blue) and diapir (red) partition coefficients for several initial perturbation radii.**
**b) Ratio of maximum fluid velocity to maximum absolute solid velocity in the entire model.**