# Peer review of "Magma ascent mechanisms in the transition regime from solitary porosity waves to diapirism"

_Solid Earth, 2020_

## Referee Comment (RC1) · Anonymous Referee #1 · 6 Aug 2020

The submitted manuscript presents parametric study of porosity wave propagation in viscous porous rocks. The novel aspect of the manuscript is the investigation of the effect of compaction length on the evolution of rising porosity waves. This is a welcome contribution since influence of material parameters and the size/geometry of the source region remains unclear. However, paper has several major drawbacks that need to be addressed.

Authors claim that they consider transition from porosity waves to diapirism. Here, I see a major conceptual problem. As often in geosciences, different terms got confused and mixed up. As I could grasp from the text, by diapirs authors understand wide

structures, while porosity waves are assumed to be narrow structures. This is already in contradiction with e.g. Wikipedia's definition of diapir, which reads as "A diapir,... is a type of geologic intrusion in which a more mobile and ductily deformable material is forced into brittle overlying rocks. Depending on the tectonic environment, diapirs can range from idealized mushroom-shaped Rayleight-Taylor-instability-type structures in regions with low tectonic stress such as in the Gulf of Mexico to narrow dykes of material that move along tectonically induced fractures in surrounding rock." Thus, according to Wikipedia all structures produced by the authors would fall into diapir category.

In the introduction authors describe diapirs as structures that are formed by Rayleigh-Taylor instability, which is commonly considered to be due to interaction of two immiscible fluids, whose behavior is described by Navier-Stokes equations. Porosity wave instability is described by Darcy law in combination with Navier-Stokes for solid. In other words, these are two different systems of equations. However, authors solve only porosity wave system of equations and thus Rayleigh-Taylor instability is not even considered in the paper. This is all very confusing for the reader and needs sharpening of the introduction and model description section. I would even suggest changing the title as diapirs in the sense of Rayleigh-Taylor instability are not even considered in the manuscript. I would suggest something more to the point, like "The effect of compaction length on solitary porosity waves and its implications for magma ascent mechanisms".

Another problem of the paper is the reliability of the presented simulation results. When changing the compaction length, authors produce porosity waves of different radius. Eventually, they become very narrow. We know from previously published research that numerical codes treating porosity waves are very sensitive to the resolution, so that several grid points are required for accurate results [Rass et al., 2019]. Thus, convergence of numerical results at higher resolution needs to be checked before acceptance of the paper. This is especially important for r'>10. We see from results

presented in the first row of Figure 1 (low values of r') that porosity waves are circular blobs as expected. Other results exhibit some tails below the circular wave that authors interpret as flow focusing. However, these are exactly the results that may suffer from lack of resolution. Besides, tails behind the major porosity wave were repeatedly reported from 1D and 2D numerical models [Connolly and Podladchikov, 1998; 2000; Rass et al., 2019]. These disappear when simulations are left for longer time periods and waves and allowed to propagate further from the source region. I expect that if authors will allow their waves to run longer, they will see that eventually perfectly circle blobs detach from the cloud. Thus, observed pattern is not a flow focusing as such but just an initial smearing of the fluid propagation front. Eventually secondary waves could form from the remaining cloud.

Some detailed comments:

Section 2.1. The described above possible confusion with terminology requires extra care when describing your governing equations. You really need to explain what the similarities and differences in the description of both instabilities are and what exactly is included into your equations. Please describe here underlying assumptions of the model of Dohmen et al. What kind of simplifications assumed in this model? I think that a very brief approach of referring to Dohmen et al. is inappropriate here.

Lines 50-55. List of principal notations would help the reader, given that you have a lot of quantities with complicated indexes, such as $\delta_{c0}$. Why not just $\delta$? Why Darcy velocity has complicated index $v_{sc0}$, why not just $v_D$? Why permeability has index $k_\phi$ and not just k? Are you using k for something else? Please consider carefully, how to make notations simpler.

Equation 5. It is a bit odd to see $\rho_s$ as an independent scale here together with 3 other scales (for length, velocity and viscosity). In principle, you can have only 3 independent scales in this problem. When you use them, you'll just get some non-dimensional parameters such as sedimentation rate in your system of equations.

[Figure]

Line 73. Please discuss small fluid viscosity limit. What are the typical viscosity values for solid magmatic rocks and for melt? What effects your simplified equations ignore?

Equation 11. Please comment here whether eqn (11) is a consequence of a usual Darcy equation or it follows some other governing law, e.g. Navier-Stokes? Which terms are omitted/presented?

Lines 85 - 90. You do not vary the radius of anomaly. The radius of your anomaly has always the same size. In the non-dimensional world, it is always w'=0.05L'. In the dimensional world it is always w=0.05L. What you are really looking at is the effect of lighter/heavier fluid in a more/less permeable rock, which will naturally have porosity waves of different size The description given in this para is very confusing.

Lines 90-91. Please comment how many grid points you have for the thinnest porosity wave.

Equation 14. Please explain this equation or provide reference for it.

Line 102. "As this radius and the maximum melt fraction change strongly during the run of a model" This just indicates that you did not reach steady-state wave propagation. See comment above.

Lines 105-107. I do not understand what you are trying to say here.

Section 3.1. This definition is very arbitrary. You do not have any diapirs in your model. You only have porosity waves of varying width. As we know, the speed of porosity wave depends on it size and thus you would have bigger and smaller waves travelling with different speed. It is interesting to compare those to the speed of diapirs, but they do not become diapirs here.

Line 109. "The transition from porosity wave to diapirism: Varying the initial wave radius" You do not vary initial wave radius, only compaction length, which is different.

Line 114. It is too early to talk about focusing at this depth. Your waves will become

circular when they will propagate higher.

Lines 115-125. Porosity waves are very sensitive to resolution. How many grid points do you have per porosity wave for your runs at r'>=20? All discussions for these runs are meaningless as you clearly run into a problem of not resolving a physical process properly. For all figures with r'>=20 you need to show convergence at higher resolution.

Lines 128-134. What is the point of giving analytical cases that do not correspond your simulations? You have only n=3 and m=1. All these extra cases and lines only confuse reader without much useful information.

Line 133. Again, here I see a big issue with terminology and conceptual understanding. You do not have diapirs. Porosity within your model is never higher than 6 times the background, which is 0.5

Lines 170-175. I do not see how this is relevant for your simulations and porosity waves. It is precisely the difference in solid and fluid densities that drives evolution of porosity waves.

Line 233. "This could lead to the propagation of magma-filled cracks" Again, remember that max porosity in your simulations is 3

Lines 235-236. "But this effect might not be strong enough to lead" Which effect? Considered in your manuscript or in the paper of Connolly and Podladchikov? Unclear sentence.

Lines 238-239. Did you perform simulations with varying porosity/permeability or is this a hypothetical scenario you are describing? Please refer to simulations with varying/layered media.

Figure 2. Explain whether the colored lined are obtained from your numerical simulations or equations (14) - (15). You also need to provide somewhere equation used for dashed lines and comment on the parameters used in this equation.

Figure 3. It is a very interesting idea to compare Stokes and porosity wave velocities. This is one of the central points of this study and therefore much more careful description is needed here. Which equations and which parameters did you use for both? What is the sensitivity of these equations to parameters that are kept fixed (e.g., n or $\phi_0$, etc). Obviously, your model did not reproduce any of the analytical velocities. Given the issue of resolution described above, you need to confirm your results at higher resolution. Letters on this figure and figure 4 are unreadable. Please increase the font.

References:

Connolly, J. A. D., and Y. Y. Podladchikov (1998), Compaction-driven fluid flow in viscoelastic rock, Geodin Acta, 11(2-3), 55-84, doi:Doi 10.1016/S0985-3111(98)80006-5.

Connolly, J. A. D., and Y. Y. Podladchikov (2000), Temperature-dependent viscoelastic compaction and compartmentalization in sedimentary basins, Tectonophysics, 324(3), 137-168.

Rass, L., T. Duretz, and Y. Y. Podladchikov (2019), Resolving hydromechanical coupling in two and three dimensions: spontaneous channelling of porous fluids owing to decompaction weakening, Geophys J Int, 218(3), 1591-1616, doi:10.1093/gji/ggz239.

---

## Referee Comment (RC2) · Anonymous Referee #2 · 9 Sep 2020

**General comments**

The submitted manuscript systematically investigates magma ascent dynamics in order to capture the transition from the solitary wave regime to diapirism. The authors explore this transition by varying the relative compaction length of the system - here by changing the model extend dimensions while keeping the compaction length constant. Investigating fluid transport mechanisms in Earth subsurface is of broad interest with applications not only limited to melt in the crust, and thus the study is a welcome contribution. Although the title and abstract sound promising, the study presents several important issues that need to be addressed before to be further considered for

publication.

1. The study's design The authors claim to resolve the transition from solitary wave of porosity to Stokes-like diapiric rise of magma. These two regimes are very different. The solitary waves of porosity occur in two-phase medium, when the fluid has a relative velocity compared to the solid. The diapiric ascent occurs if the fluid has no or very limited mobility with respect to the solid and thus the medium behaves as single-phase. The authors report here briefly the two-phase flow equations they rely on, which permit to resolve the two-phase motion. However, it is unclear what happens in the single-phase flow limit. In this limit, the equations should reduce to the single phase (Navier-) Stokes system. This part is totally absent from the study, both in the physical description (system of equations) and from the numerical implementation. The authors overlooked a study from Scott (1988) investigating a very similar research question, namely "The competition between percolation and circulation in a deformable porous medium". This short communication may be highly relevant and may support or challenge some statement claimed by the authors.

2. The numerical implementation In this study, the authors rely on numerical modelling to investigates the effect of changes in compaction length, or rather vary the domain size keeping the compaction length fixed. Being a numerical study, the current manuscript seriously lacks in robust model description, numerical implementation, benchmarking. These (non-exhaustive) steps are the basic technicalities one is expected to report when performing numerical experiments. The authors emphasise both in the Abstract and the Introduction the numerical challenges relative to accurately resolving fluid migration in the subsurface. However, no further discussion about numerical method, implementation, benchmarking, sensitivity analysis, etc... is present in the manuscript. The model configuration is poorly described and some basic information such as the numerical grid resolution should be reported in a well-crafted "Numerical Implementation" section well before the final discussion. Although focus should not be on benchmarking, ensuring accuracy of the numerical scheme and related results is primordial in studies like this one. As reported recently by Räss et al. (2019), lack of numerical resolution may lead to erroneous results. I am afraid that part of the results reported in this study are under-resolved, as at least a few tens of grid-points are needed per compaction length to obtain accurate results. Also missing is the description of the transition from two-phase flow to single-phase flow. How do the authors treat the very small compaction length limit? In this limit, Stokes flow is dominating, and the motion of the fluid pocket needs advection of the solid matrix. There is no information regarding this important point in the manuscript. The governing equations are very cryptic, and it would be very helpful to see the finally implemented closed system of equations that is actually solved numerically, together with information on the numerical scheme that is used.

3. The quality of the reported results The reported results are interesting but in light of the previous comments, further work would be welcome to refine the Results and Discussion sections. The authors could put some additional efforts in crafting better quality figures. There are missing labels, fonts are very small and hardly readable in some cases, and figure captions display repetitions and miss important details. Also, it may be interesting to report in form of quiver plots the solid and fluid velocity components as those could be directly compared to results obtained by Scott in 1988.

To summarise, this manuscript tackles an interesting and not yet fully resolved question, but the study's design, numerical implementation and overall quality should be seriously improved before being considered for publication. Addressing these issues are important as in the current status it is hard for the reader to discriminate between resolved dynamics or numerical artefacts, especially in the transition regime. In the Discussion, the authors provide some insights in the challenges related to resolving the two-phase dynamics for large domains (or small compaction length). There may be a conceptual study design issue there. The authors spell out all the pitfall and they don't, but their study actually reports results that exactly suffer from those drawbacks. and may not be accurate. A potential way to improve the study would be to move a

large part of the issues raised in the discussion to the Section 2. For example, the discussion about the numerical grid resolution should appear much earlier. Then, one could discuss the issue, try to solve it. And if results cannot be trusted, then one should identify them and discard them from the analysis.

**Detailed comments**

l.18: In the current status, these may be numerical artefacts as well. Appropriate benchmarking would be welcome (e.g. running a test setup at various resolutions and reporting the results).

l.21: For accurate results of porosity waves, numerical resolution should always be such to have about 10 grid points per compaction length.

l.23-24: True, one should be careful. Please report how you carefully addressed these resolution issues.

l.47-49: Important question on "what are the numerical implications on modelling magma transport". Within the manuscript, however, these implications are discussed but it appears that the suggestions provided are not followed by the authors themselves.

Section Introduction: Please update it putting your contribution in light of previous work such as Scott (1988) and other potential studies.

eq.8-11: These are non-intuitive formulation of the momentum balance. What do $v_1$ and $v_2$ stand for? Please take some place to better describe the approach.

Section 2.2: Please complete the model setup.

l.85: What value of A do you use in the experiments?

l.90: This may be problematic as number of grid points per compaction length will decrease with increased nondimensional box size.

l.92: Can you precise what out and inflow conditions you use for the solid? Majority two-phase flow simulation apply free slip boundary conditions for the solid or porous matrix. Please clarify the model configuration - this is crucial for reproducible science.

l.93: What do mirroring boundary conditions refer to?

l.98 + eq.14: Please provide relevant reference for the Stokes velocity?

l.99: Please justify the choice of the radius you utilise in the Stokes formula.

eq.15: Ar not defined

Section 2: Besides the model setup, please report what final equations are implemented in the numerical model. Please also report about your numerical implementation, discretisation, solution strategy; all standard components one is expected to see in a numerical study that would enable reproducible science.

l.117: This may indeed show lack of numerical resolution.

l.121-126: No focussing is expected for linear shear and bulk rheology. The focussing you report here may rather be attributed to the still transient state of the model evolution - maybe due to the coarse resolution. To verify this, a higher resolution simulation on a larger domain should be carried out and running until the shape stabilises.

l.128-132: Why to report various analytical values when your simulation was carried out only with n=3, m=1. This only confuses the reader.

l.164.167: Internal circulation would be great to see in a figure. It is difficult to assess and acknowledge your findings based on text only.

l.170-171: How can you neglect the density difference between solid and melt. This should be the driving force.

l.218: This conclusion should be verified by a higher resolution run.

Section 3: May need further development upon updated results

l.224-248: Interesting insight but all these hypotheses should be tested within appropriate modelling framework including spatial variations in the suggested material parameter fields and using sufficient numerical grid resolution to allow resolving the smallest features. Also, note that focussing will only occur if there is asymmetry among compaction and decompaction of the porous matrix, i.e. for non-linear rheology.

l.249-264: Good point, but it seems that this study exactly shows the reported artefacts in the results.

**References**

Scott, D. R. (1988). The competition between percolation and circulation in a deformable porous medium. Journal of Geophysical Research: Solid Earth, 93(B6), 6451-6462.

Räss, L., Duretz, T., & Podladchikov, Y. Y. (2019). Resolving hydromechanical coupling in two and three dimensions: spontaneous channelling of porous fluids owing to decompaction weakening. Geophysical Journal International, 218(3), 1591-1616.

---

## Author Comment (AC1) · 23 Oct 2020

Thanks to the reviewers we were able to greatly improve this manuscript and we could hopefully answer all comments made good enough. It was quite a lot of work we put into this manuscript after receiving the reviews and a big part of it was completely rewritten or changed and quoting it all here in the answer letter would mage it hard to follow. Therefore, I will just briefly explain the biggest changes made here and kindly refer to the new manuscript.

We completely rewrote the mathematical description of the model which now includes the dimensional equations and a small chapter about how these equations are solved. We also use a new non-dimensionalization using the Stokes velocity and the radius of the initial perturbation. This allows us now to describe the Stokes limit, where the old description failed. It might also help to better understand the quite complex model setup. Due to the change in scaling all figures had to be remade and are now hopefully up to the standards and everything is readable.

Regarding the figures, we removed former figure 3 as we think that it didn't give much more information than figure 2 already gave and is quite complicated to understand and describe. Instead we added a new figure containing a resolution test which is described and analyzed in a new chapter called "numerical issues".

The main point in this new manuscript is that we now state that the focusing, we formerly stated are small porosity waves, are channels that build up in front of the wave due to the horizontal stresses occurring there. This new statement is described and analyzed in the results part of the manuscript, where it replaces the argument of the small porosity waves.

In the discussion we now discuss the growth rates of these channels in our models and compare them to Stevenson (1989).

Several other parts in the manuscript had to be changed according to our new statement and are not especially mentioned here but are marked in the updated manuscript in red.

Below you will find the comments made by the reviewers in black and our answers in red.

**Anonymous Referee #1**

*The submitted manuscript presents parametric study of porosity wave propagation in viscous porous rocks. The novel aspect of the manuscript is the investigation of the effect of compaction length on the evolution of rising porosity waves. This is a welcome contribution since influence of material parameters and the size/geometry of the source region remains unclear. However, paper has several major drawbacks that need to be addressed.*

*Authors claim that they consider transition from porosity waves to diapirism. Here, I see a major conceptual problem. As often in geosciences, different terms got confused and mixed up. As I could grasp from the text, by diapirs authors understand wide structures, while porosity waves are assumed to be narrow structures. This is already in contradiction with e.g. Wikipedia's definition of diapir, which reads as "A diapir,. . . is a type of geologic intrusion in which a more mobile and ductily deformable material is forced into brittle overlying rocks. Depending on the tectonic environment, diapirs can range from idealized mushroom-shaped Rayleight-Taylor-instability-type structures in regions with low tectonic stress such as in the Gulf of Mexico to narrow dykes of material that move along tectonically induced fractures in surrounding rock." Thus, according to Wikipedia all structures produced by the authors would fall into diapir category.*

We do not agree with the English version of Wikipedia. Diapirism is not necessarily related to brittle overburden and to a more mobile buoyant material. We use diapirism in the sense it had been defined and introduced e.g. by Turcotte and Schubert (1981), Simpson, 1989, and many others in the 1970s and 1980s. We specify our (and the common geologic) definition:

"Addressing different melt ascent mechanisms it may be useful to specify our definition of diapirism. Originating from the Greek "*diapeirein*", i.e. "to pierce through", diapirism describes the "buoyant upwelling of relatively light rock" (Turcotte and Schubert, 1981) through and into a denser overburden. In the general definition the rheology of the diapir and ambient material is not specified, both can be ductile as in our case, but often, the overburden is assumed being more viscous or even brittle. Buoyancy may be of compositional or phase related origin, e.g. due to the presence of non-segregating partial melt (Wilson, 1989). Based on these definitions in our case a diapir is a rising, partially molten body or porosity anomaly with zero fluid-solid separation velocity. Mathematically the equations of motion of the two-phase system degenerate to the Stokes equation (see below)."

*In the introduction authors describe diapirs as structures that are formed by RayleighTaylor instability, which is commonly considered to be due to interaction of two immiscible fluids, whose behavior is described by Navier-Stokes equations. Porosity wave instability is described by Darcy law in combination with Navier-Stokes for solid. In other words, these are two different systems of equations. However, authors solve only porosity wave system of equations and thus Rayleigh-Taylor instability is not even considered in the paper. This is all very confusing for the reader and needs sharpening of the introduction and model description section. I would even suggest changing the title as diapirs in the sense of Rayleigh-Taylor instability are not even considered in the manuscript. I would suggest something more to the point, like "The effect of compaction length on solitary porosity waves and its implications for magma ascent mechanisms".*

We do not consider or mention Rayleigh Taylor instability in our paper, per definition a RT instability starts from a stratified and not from a buoyant circular anomaly. But thanks to the reviewer, we specify the Darcy and Stokes type of equations for the both end member now.

*Another problem of the paper is the reliability of the presented simulation results. When changing the compaction length, authors produce porosity waves of different radius. Eventually, they become very narrow. We know from previously published research that numerical codes treating porosity waves are very sensitive to the resolution, so that several grid points are required for accurate results [Rass et al., 2019]. Thus, convergence of numerical results at higher resolution needs to be checked before acceptance of the paper. This is especially important for r'>10. We see from results presented in the first row of Figure 1 (low values of r') that porosity waves are circular blobs as expected. Other results exhibit some tails below the circular wave that authors interpret as flow focusing. However, these are exactly the results that may suffer from lack of resolution. Besides, tails behind the major porosity wave were repeatedly reported from 1D and 2D numerical models [Connolly and Podladchikov, 1998; 2000; Rass et al., 2019]. These disappear when simulations are left for longer time periods and waves and allowed to propagate further from the source region. I expect that if authors will allow their waves to run longer, they will see that eventually perfectly circle blobs detach from the cloud. Thus, observed pattern is not a flow focusing as such but just an initial smearing of the fluid propagation front. Eventually secondary waves could form from the remaining cloud.*

Yes, numerical resolution is a major problem in modelling porosity waves and or model setup will inevitably lack in decent compaction length resolution. Anyways, we now state that the peaks we observe in the transitional regime are channels which are still resolvable with our resolution. Channels also explain why the tail behind the leading wave does not get smaller or even vanish but growth with time. We now calculate growth rates and show that they agree with Stevenson (1989).

*Still, resolution is a major issue and we added a new chapter about "numerical issues".*

*Some detailed comments:*

*Section 2.1. The described above possible confusion with terminology requires extra care when describing your governing equations. You really need to explain what the similarities and differences in the description of both instabilities are and what exactly is included into your equations. Please describe here underlying assumptions of the model of Dohmen et al. What kind of simplifications assumed in this model? I think that a very brief approach of referring to Dohmen et al. is inappropriate here.*

Now we specify how the former equ 11 is derived, which reveals the inherent assumptions. As for the "small fluid viscosity limit" we add:

"In the small fluid viscosity limit the viscous stresses within the fluid phase are neglected, resulting in a viscous stress tensor in the Stokes equation of the mixture (equ. 4), in which only the stresses in the solid phase are relevant. This is evident from the definition of the viscous stress tensor, which only contains matrix and not fluid viscosities. Melt viscosities of carbonatitic, basaltic or silicic wet or dry melts span a range from < 1 Pa s to extreme values up to $10^{14}$ Pa s (see the discussion in Schmeling et al., 2019), while effective viscosities of mafic or silicic partially molten rocks may range between $10^{20}$ Pa s and $10^{16}$ Pa s, depending on melt fraction, stress, and composition. Thus, in most circumstances the small fluid viscosity limit is justified."

*Lines 50-55. List of principal notations would help the reader, given that you have a lot of quantities with complicated indexes, such as δc0. Why not just δ? Why Darcy velocity has complicated index vsc0, why not just vD? Why permeability has index kφ and not just k? Are you using k for something else? Please consider carefully, how to make notations simpler. Equation 5. It is a bit odd to see ρs as an independent scale here together with 3 other scales (for length, velocity and viscosity). In principle, you can have only 3 independent scales in this problem. When you use them, you'll just get some non-dimensional parameters such as sedimentation rate in your system of equations.*

As we completely revised our mathematical description, this problem is hopefully solved. We still stick to $k_{\varphi}$ and $\delta_c$ as these notations are commonly used but we got rid of the zero notations.

*Line 73. Please discuss small fluid viscosity limit. What are the typical viscosity values for solid magmatic rocks and for melt? What effects your simplified equations ignore? Equation 11. Please comment here whether eqn (11) is a consequence of a usual Darcy equation or it follows some other governing law, e.g. Navier-Stokes? Which terms are omitted/presented?*

See above.

*Lines 85 - 90. You do not vary the radius of anomaly. The radius of your anomaly has always the same size. In the non-dimensional world, it is always w'=0.05L'. In the dimensional world it is always w=0.05L. What you are really looking at is the effect of lighter/heavier fluid in a more/less permeable rock, which will naturally have porosity waves of different size The description given in this para is very confusing.*

This was probably also caused due to the confusion with the non-dimensionalization. But we are in fact changing the radius of the emerging solitary wave. When we double the characteristic compaction length of the model the solitary wave will be also double the size in [m] as the wave will be the same size in terms of compaction length.

Now with the new description it might be easier to understand the model setup. We also gave a small example.

*Lines 90-91. Please comment how many grid points you have for the thinnest porosity wave.*

As we now state that they are channels and not porosity waves the number of grid points per wave is no longer that important, but we also have a look at the resolution with a new figure with different resolutions. We see that the channels are resolvable as long as the grid size is approximately in the order of the compaction length.

*Equation 14. Please explain this equation or provide reference for it.*

This equation is the commonly used Stokes equation, which is now referenced with (Turcotte & Schubert, 1981).

*Line 102. "As this radius and the maximum melt fraction change strongly during the run of a model" This just indicates that you did not reach steady-state wave propagation. See comment above.*

See above.

*Lines 105-107. I do not understand what you are trying to say here.*

Yes, this sentence was a bit odd and is no longer part of the new mathematical description.

*Section 3.1. This definition is very arbitrary. You do not have any diapirs in your model. You only have porosity waves of varying width. As we know, the speed of porosity wave depends on it size and thus you would have bigger and smaller waves travelling with different speed. It is interesting to compare those to the speed of diapirs, but they do not become diapirs here.*

This might be a confusion due to the old description of the theory, as it wasn't able to describe the Stokes limit. With the new description we now are able to describe it and it might be now clear that we get diapirs in the sense we stated above.

*Line 109. "The transition from porosity wave to diapirism: Varying the initial wave radius" You do not vary initial wave radius, only compaction length, which is different.*

See above.

*Line 114. It is too early to talk about focusing at this depth. Your waves will become circular when they will propagate higher.*

Now with the channels we clearly observe focusing in the sense that melt gets accumulated in a smaller horizontal area.

*Lines 115-125. Porosity waves are very sensitive to resolution. How many grid points do you have per porosity wave for your runs at r'>=20? All discussions for these runs are meaningless as you clearly run into a problem of not resolving a physical process properly. For all figures with r'>=20 you need to show convergence at higher resolution.*

*See above.*

*Lines 128-134. What is the point of giving analytical cases that do not correspond your simulations? You have only n=3 and m=1. All these extra cases and lines only confuse reader without much useful information.*

*The figure has been revised and now shows just the relevant case of n=3 and m=1.*

*Line 133. Again, here I see a big issue with terminology and conceptual understanding. You do not have diapirs. Porosity within your model is never higher than 6 times the background, which is 0.5*

*See above.*

*Lines 170-175. I do not see how this is relevant for your simulations and porosity waves. It is precisely the difference in solid and fluid densities that drives evolution of porosity waves.*

This is part of our Boussinesq approximation, where all density differences are dropped but in the buoyancy term of the momentum equations.

We now describe this approximation in our mathematical description.

*Line 233. "This could lead to the propagation of magma-filled cracks" Again, remember that max porosity in your simulations is 3*

Yes, this is true, but the melt porosities would be way higher if we would allow for further focusing. Also our model starts with very low melt porosities as these models are more robust, but we would observe the same behavior for higher porosities.

*Lines 235-236. "But this effect might not be strong enough to lead" Which effect? Considered in your manuscript or in the paper of Connolly and Podladchikov? Unclear sentence.*

The sentence has been changed to:

"But this upward weakening might not be strong enough to lead to the focusing needed for the nucleation of dykes"

*Lines 238-239. Did you perform simulations with varying porosity/permeability or is this a hypothetical scenario you are describing? Please refer to simulations with varying/layered media.*

We did some simple tests with several different layers that have different shear and bulk viscosities and solitary waves passing through them as port of another project. This test was just very simple and not enough to show in this paper but shows exactly what we describe here, a focusing. We did not, however, perform simulations with varying background porosity as this model setup might be even more complex. The case of differing viscosity should however have the same effect, as it also changes the compaction length.

The sentence has been changed to:

"In the hypothetic case of a porosity wave reaching the top of a magma chamber, the background porosity might decrease which would most certainly lead to focusing, because the compaction length will decrease, and eventually, when reaching melt free rocks, the melt rich fingers may stall as in our models at $r > 50 \cdot \delta_c$ and the rising melt will accumulate and enter the pure diapirism regime"

*Figure 2. Explain whether the colored lined are obtained from your numerical simulations or equations (14) - (15). You also need to provide somewhere equation used for dashed lines and comment on the parameters used in this equation.*

The dashed lines in the original figure are now the colored ones. They are calculated semi-analytically using the program provided by Simpson & Spiegelman (2011) and there is therefore no analytical equation to describe these curves.

The former colored lines were calculated analytically with the mentioned equation for the Stokes sphere, but this information is now no longer needed.

The caption of the figure was changed to:

The dashed line marks the velocity of the Stokes sphere ($v' = 1$). The colored lines show the velocity of a 2D solitary wave, calculated semi-analytically by Simpson & Spiegelman (2011), in our non-dimensionalization, based on the radii shown in the legend.

*Figure 3. It is a very interesting idea to compare Stokes and porosity wave velocities. This is one of the central points of this study and therefore much more careful description is needed here. Which equations and which parameters did you use for both? What is the sensitivity of these equations to parameters that are kept fixed (e.g., n or φ0, etc). Obviously, your model did not reproduce any of the analytical velocities. Given the issue of resolution described above, you need to confirm your results at higher resolution. Letters on this figure and figure 4 are unreadable. Please increase the font.*

We now got rid of figure 3 as it was quite complex and did not really give any more information that is not already in figure 2. Sensitivity to the parameters kept fixed is a whole different story. Changing $\varphi_0$ should lead to minor changes in the results as we used simplified viscosities. In Dohmen et al. (2019) we have a look at the behavior of SWs for different background porosities. They play a major roll with the more complex, lower viscosities, used there. Changing *n* would probably change the results, but it would need much more time to get a similar work with n=2.

References:

Connolly, J. A. D., and Y. Y. Podladchikov (1998), Compaction-driven fluid flow in viscoelastic rock, Geodin Acta, 11(2-3), 55-84, doi:Doi 10.1016/S0985-3111(98)80006-5.

Connolly, J. A. D., and Y. Y. Podladchikov (2000), Temperature-dependent viscoelastic compaction and compartmentalization in sedimentary basins, Tectonophysics, 324(3), 137-168.

Rass, L., T. Duretz, and Y. Y. Podladchikov (2019), Resolving hydromechanical coupling in two and three dimensions: spontaneous channelling of porous fluids owing to decompaction weakening, Geophys J Int, 218(3), 1591-1616, doi:10.1093/gji/ggz239.

---

## Author Comment (AC2) · 23 Oct 2020

Thanks to the reviewers we were able to greatly improve this manuscript and we could hopefully answer all comments made good enough. It was quite a lot of work we put into this manuscript after receiving the reviews and a big part of it was completely rewritten or changed and quoting it all here in the answer letter would mage it hard to follow. Therefore, I will just briefly explain the biggest changes made here and kindly refer to the new manuscript.

We completely rewrote the mathematical description of the model which now includes the dimensional equations and a small chapter about how these equations are solved. We also use a new non-dimensionalization using the Stokes velocity and the radius of the initial perturbation. This allows us now to describe the Stokes limit, where the old description failed. It might also help to better understand the quite complex model setup. Due to the change in scaling all figures had to be remade and are now hopefully up to the standards and everything is readable.

Regarding the figures, we removed former figure 3 as we think that it didn't give much more information than figure 2 already gave and is quite complicated to understand and describe. Instead we added a new figure containing a resolution test which is described and analyzed in a new chapter called "numerical issues".

The main point in this new manuscript is that we now state that the focusing, we formerly stated are small porosity waves, are channels that build up in front of the wave due to the horizontal stresses occurring there. This new statement is described and analyzed in the results part of the manuscript, where it replaces the argument of the small porosity waves.

In the discussion we now discuss the growth rates of these channels in our models and compare them to Stevenson (1989).

Several other parts in the manuscript had to be changed according to our new statement and are not especially mentioned here but are marked in the updated manuscript in red.

Below you will find the comments made by the reviewers in black and our answers in red.

***Anonymous Referee #2***

**General comments**

The submitted manuscript systematically investigates magma ascent dynamics in order to capture the transition from the solitary wave regime to diapirism. The authors explore this transition by varying the relative compaction length of the system - here by changing the model extend dimensions while keeping the compaction length constant. Investigating fluid transport mechanisms in Earth subsurface is of broad interest with applications not only limited to melt in the crust, and thus the study is a welcome contribution. Although the title and abstract sound promising, the study presents several important issues that need to be addressed before to be further considered for publication.

1. The study's design

The authors claim to resolve the transition from solitary wave of porosity to Stokes-like diapiric rise of magma. These two regimes are very different. The solitary waves of porosity occur in two-phase medium, when the fluid has a relative velocity compared to the solid. The diapiric ascent occurs if the fluid has no or very limited mobility with respect to the solid and thus the medium behaves as single-phase. The authors report here briefly the two-phase flow equations they rely on, which permit to resolve the two-phase motion. However, it is unclear what happens in the single-phase flow limit. In this limit, the equations should reduce to the single phase (Navier-) Stokes system. This part is totally

absent from the study, both in the physical description (system of equations) and from the numerical implementation. The authors overlooked a study from Scott (1988) investigating a very similar research question, namely "The competition between percolation and circulation in a deformable porous medium". This short communication may be highly relevant and may support or challenge some statement claimed by the authors.

We totally agree with the reviewer here. The equations given in the former version describe the two-phase flow limit but fail in the Stokes limit. Because of that we introduced a new non-dimensionalization that is capable of describing both limits.

The "Governing equations" section was completely rewritten.

We now mention the research of Scott (1988):

This switch from negative to positive mass flux was already observed by Scott (1988), but while he changed the viscosity ratio, we change the radius and keep the viscosity ratio constant. Both describe the transition from a two-phase limit towards the Stokes limit, but in our formulation we are able to reach the Stokes limit while Scott (1988) is still in the two-phase flow regime.

2. The numerical implementation

In this study, the authors rely on numerical modelling to investigates the effect of changes in compaction length, or rather vary the domain size keeping the compaction length fixed. Being a numerical study, the current manuscript seriously lacks in robust model description, numerical implementation, benchmarking. These (non-exhaustive) steps are the basic technicalities one is expected to report when performing numerical experiments. The authors emphasise both in the Abstract and the Introduction the numerical challenges relative to accurately resolving fluid migration in the subsurface. However, no further discussion about numerical method, implementation, benchmarking, sensitivity analysis, etc... is present in the manuscript. The model configuration is poorly described and some basic information such as the numerical grid resolution should be reported in a well-crafted "Numerical Implementation" section well before the final discussion. Although focus should not be on benchmarking, ensuring accuracy of the numerical scheme and related results is primordial in studies like this one. As reported recently by Räss et al. (2019), lack of numerical resolution may lead to erroneous results. I am afraid that part of the results reported in this study are under-resolved, as at least a few tens of gridpoints are needed per compaction length to obtain accurate results. Also missing is the description of the transition from two-phase flow to single-phase flow. How do the authors treat the very small compaction length limit? In this limit, Stokes flow is dominating, and the motion of the fluid pocket needs advection of the solid matrix. There is no information regarding this important point in the manuscript. The governing equations are very cryptic, and it would be very helpful to see the finally implemented closed system of equations that is actually solved numerically, together with information on the numerical scheme that is used.

Yes, the numerical resolution is a major issue, but now, as we revised our statements, the resolution is no longer as big of a problem as before. The small porosity waves we observed in the transition regime would have been most certainly not decently resolve, but now we state that we observe channeling in this regime, based on Stevenson (1989), which are resolvable by our resolution. The channel's wavelength in our models is in the same order as in Stevenson (1989) and the growth rate is explainable as well.

We still added a small chapter about numerical issues to the results, that tells a little bit about the issues observed.

We also give a small introduction on how we solve the equations numerically and, as already stated above, we changed the mathematical description so that we are now able to reach the Stokes Limit.

3. The quality of the reported results

The reported results are interesting but in light of the previous comments, further work would be welcome to refine the Results and Discussion sections. The authors could put some additional efforts in crafting better quality figures. There are missing labels, fonts are very small and hardly readable in some cases, and figure captions display repetitions and miss important details. Also, it may be interesting to report in form of quiver plots the solid and fluid velocity components as those could be directly compared to results obtained by Scott in 1988.

All figures were revised and are now hopefully up to the standards.

To summarise, this manuscript tackles an interesting and not yet fully resolved question, but the study's design, numerical implementation and overall quality should be seriously improved before being considered for publication. Addressing these issues are important as in the current status it is hard for the reader to discriminate between resolved dynamics or numerical artefacts, especially in the transition regime. In the Discussion, the authors provide some insights in the challenges related to resolving the two-phase dynamics for large domains (or small compaction length). There may be a conceptual study design issue there. The authors spell out all the pitfall and they don't, but their study actually reports results that exactly suffer from those drawbacks. and may not be accurate. A potential way to improve the study would be to move a large part of the issues raised in the discussion to the Section 2. For example, the discussion about the numerical grid resolution should appear much earlier. Then, one could discuss the issue, try to solve it. And if results cannot be trusted, then one should identify them and discard them from the analysis.

**Detailed comments**

l.18: In the current status, these may be numerical artefacts as well. Appropriate benchmarking would be welcome (e.g. running a test setup at various resolutions and reporting the results).

We now state that the "numerical artefacts" mentioned are channels which are resolvable and show its dependence of resolution in a resolution test.

l.21: For accurate results of porosity waves, numerical resolution should always be such to have about 10 grid points per compaction length.

Yes, such a resolution would be desirable, but is hard to reach in many models. Anyways, as we now state channels this minimum resolution criteria is no longer applicable.

l.23-24: True, one should be careful. Please report how you carefully addressed these resolution issues.

See above.

l.47-49: Important question on "what are the numerical implications on modelling magma transport". Within the manuscript, however, these implications are discussed but it appears that the suggestions provided are not followed by the authors themselves.

See above.

Section Introduction: Please update it putting your contribution in light of previous work such as Scott (1988) and other potential studies.

We added a small comparison of our models to Scott (1988):

*"Scott (1988) already had a look at a similar scenario. He calculated porosity waves changing the compaction length by altering the shear to bulk viscosity ratio, while we want to change the radius of a partially molten perturbation in terms of compaction lengths but keeping the viscosity constant. While Scott (1988) was not able to reach the single-phase flow endmember due to his setup we can reach this endmember with our description and can show how the transition looks like."*

eq.8-11: These are non-intuitive formulation of the momentum balance. What do v_1 and v_2 stand for? Please take some place to better describe the approach.

We completely revised the mathematical description and now explain how we get the momentum balance. This description is hopefully more intuitive. v_1 and v_2 have been explained as well.

Section 2.2: Please complete the model setup.

With the new non-dimensionalization the model setup is hopefully better understandable. We also give a small example as the model series is not really intuitive.

l.85: What value of A do you use in the experiments?

We now mention the Amplitude of the wave:

" … where $A$ is the amplitude equal to 0.03 in our models…"

l.90: This may be problematic as number of grid points per compaction length will decrease with increased nondimensional box size.

Yes, this is a problem, even though we now observe channels. But it is not really possible to keep the resolution of the compaction length constant. From r'=1.5 to r'=100 we would have to increase the resolution with a factor of 66, corresponding to a resolution of 13201x13201. Even when we say we don't need a higher resolution for the bigger radii as the compaction length doesn't need to be resolved as good, we still have very high resolutions with high CPU-times. We also observed that some models become unstable with very high resolutions, which is not explainable by now.

l.92: Can you precise what out and inflow conditions you use for the solid? Majority two-phase flow simulation apply free slip boundary conditions for the solid or porous matrix. Please clarify the model configuration - this is crucial for reproducible science.

We now describe the in and outflow more:

"At the top and the bottom, we prescribe an out- and inflow for both melt and solid, respectively, which is calculated analytically for the background porosity. This is necessary because we have a background melt fraction $\varphi_0$, that has a certain buoyancy which would lead to an accumulation of melt at the top of the model. We therefore calculate the segregation velocity for background porosity using equation (17) without the viscous stress term. The corresponding matrix velocity is calculated using the conservation of mass."

l.93: What do mirroring boundary conditions refer to?

We now explain the mirroring boundary conditions:

"At the sides we use mirroring boundary conditions, which corresponds to a symmetry axis, where no horizontal flow is allowed."

l.98 + eq.14: Please provide relevant reference for the Stokes velocity?

The Stokes velocity is now introduced earlier in the mathematical description and a reference has been added: Turcotte & Schubert (1982).

l.99: Please justify the choice of the radius you utilise in the Stokes formula.

We added a small justification:

"We use the halfwidth of the initial perturbation as radius for the Stokes velocity. This is reasonable as the amount of melt in the perturbation is approximately equal to the amount of melt in a spheres cut with a sharp boundary of radius $r$, for what the Stokes equation is valid."

eq.15: Ar not defined

Ar was actually A times r, where A is the amplitude of the initial perturbation and r its radius. With the new description A has been replaced by $\varphi_{max}$.

Section 2: Besides the model setup, please report what final equations are implemented in the numerical model. Please also report about your numerical implementation, discretisation, solution strategy; all standard components one is expected to see in a numerical study that would enable reproducible science.

The new mathematical description might now solve this comment, as we now start with the dimensional equations. We also added a paragraph to the numerical strategy.

l.117: This may indeed show lack of numerical resolution.

See above.

l.121-126: No focussing is expected for linear shear and bulk rheology. The focussing you report here may rather be attributed to the still transient state of the model evolution - maybe due to the coarse resolution. To verify this, a higher resolution simulation on a larger domain should be carried out and running until the shape stabilises.

We now state that this focusing is a channel which is able to evolve with the rheology used in this work.

l.128-132: Why to report various analytical values when your simulation was carried out only with n=3, m=1. This only confuses the reader.

Good point. With the revision of the figures we now show just the n=3, m=1 case. With the new depiction it would have been even more confusing.

l.164.167: Internal circulation would be great to see in a figure. It is difficult to assess and acknowledge your findings based on text only.

As this chapter was deleted, we don't mention the internal circulation. Just for interest one could add a vector field to one of the figures, but the waves shown are all to small to see something then. Adding a new figure wouldn't make much sense as it wouldn't be referred to.

l.170-171: How can you neglect the density difference between solid and melt. This should be the driving force.

We neglect the density difference everywhere but in the buoyancy terms of the momentum equations. This is part of the Boussinesq approximation, we now explain in the mathematical description.

l.218: This conclusion should be verified by a higher resolution run.

See above.

Section 3: May need further development upon updated results

The section was partly rewritten and now addresses some of the issues stated above.

l.224-248: Interesting insight but all these hypotheses should be tested within appropriate modelling framework including spatial variations in the suggested material parameter fields and using sufficient numerical grid resolution to allow resolving the smallest features. Also, note that focussing will only occur if there is asymmetry among compaction and decompaction of the porous matrix, i.e. for non-linear rheology.

We now replaced focusing with channeling which is able to evolve with linear rheology. Still we are not able to resolve even the smallest features but the channeling we now state is less affected by the lack of numerical resolution.

l.249-264: Good point, but it seems that this study exactly shows the reported artefacts in the results.

See above.

**References**

Scott, D. R. (1988). The competition between percolation and circulation in a deformable porous medium. Journal of Geophysical Research: Solid Earth, 93(B6), 6451-6462.

Räss, L., Duretz, T., & Podladchikov, Y. Y. (2019). Resolving hydromechanical coupling in two and three dimensions: spontaneous channelling of porous fluids owing to decompaction weakening. Geophysical Journal International, 218(3), 1591-1616

---

## Referee Report (RR1)

570

[referee-annotated manuscript omitted]

---

## Referee Report (RR2)

595

[referee-annotated manuscript omitted]

---

## Author Response (AR2)

Thanks to the reviewers, we could greatly improve our manuscript and are now hopeful that all stated issues are solved to satisfaction. Most of the manuscript was completely rewritten and we therefore kindly ask the reviewers to have a look at the revised manuscript where all changes have been marked. Citing all changes in this letter would be too messy. Nevertheless, our answers are marked in red and if the changes were just slightly, we still cited them here in green.

To give a broad overview over the changes, here a small summary:

We changed the model setup in a way that the coordinate system follows the maximum melt fraction, which allowed us to zoom into the initial perturbation, which in turn helped us reaching the required resolutions for the compaction length. With this improved resolution we no longer observe channels, but solitary waves. These will build up independently of how slow the segregation velocity is, if the ascend time is high enough. For very high radii a diapir will split up into numerous solitary waves but ascend as a whole, mostly affected by the surrounding matrix.

We got rid of the retention number in our mathematical description and replaced it by the squared ratio of compaction length to model length scale r. Following this the segregation to Stokes velocity ration analysis was expanded by a figure, which shows the results of it for a few initial perturbation maxima. They fit nicely to the observed results of our models.

**Reviewer 2**

**Suggestions for revision or reasons for rejection (will be published if the paper is accepted for final publication)**

The submitted revised manuscript by Dohmen and Schmeling systematically investigates magma ascent dynamics in order to capture the transition from the solitary wave regime to diapirism. The authors explore this transition by varying the relative compaction length of the system - here by changing the model extend while keeping the compaction length constant.

The authors addressed majority of the concerns raised during the first round of revisions. However, the new version of the submitted manuscript still suffers from major design issues, both in the content and form. Rather than "time investment and good enough-ness", focus should be on scientific approach and accuracy. To the point, the main story of the manuscript -channelling- is not receivable as such. The authors motivate their revised study by unveiling apparent channelling mechanism occurring while transitioning from the solitary-wave to the Stokes regime. Their argumentation would only be receivable if following a scientific approach, i.e., including more than wishful thinking.

Simple words for simple things; Assuming there is a not yet discovered channelling mechanisms, natural steps would be following: (1) provide a parameter accounting for it; (2) test and report the influence of this parameter in a systematic study; (3) prove the robustness of the suggested results by providing (numerical) convergence tests (physical results should no longer vary with further increase in numerical resolution - proof of a robust numerical implementation) targeting the configuration of interest. To date none of these steps are successfully implemented.

Now, and unless proven otherwise, the underlying equation do not contain any channelling mechanism as such. Thus, the reported channels may rather be the expression of a lack in numerical resolution. This conclusion still confirms the outcome of previous reviews.

Channelling ultimately requires an asymmetry in compaction versus decompaction regimes, obtained upon nonlinear bulk rheology by mechanisms such as e.g., decompaction

weakening (Connolly and Podladchikov, 1998; Räss, 2018; Räss, 2019) or brittle failure (Keller, 2013; Yarushina, 2015). Moreover, including the full shear stress tensor for the mixture velocities and total pressure won't produce extra focussing and asymmetry; neither would porosity dependent and even strain-dependent shear rheology. Both may impact the compaction length which may further influence the relative inclusion size, at most.

Now, after following, the remarks about numerical solution wo do not, as supposed above, observe channeling anymore. The finger-like features turned out to be not-sufficiently resolved solitary waves. We therefore got rid of all text passages concerning channeling.

Finally, the effort spent in providing further insights into the underlying physics and mathematical model (Section 2.1) is very much appreciated. However, this new section reports inconsistent derivations. Equation (3) reports de analogy of the fluid momentum equations as a generalised Darcy law that contains de gradient of the fluid pressure P minus the buoyant fluid force $\rho fg$. Equation (4) reports the total force or momentum balance, where the viscous stresses and total (mixture) pressure P equilibrate the total buoyancy force $\bar{\rho}g$. The pressure term in equation (3) represents as such the fluid pressure Pf, while the pressure in equation (4) stands for the total pressure Ptot. Equation (10) and line 96 is thus wrong. Fluid pressure ≠ total pressure (Pf≠Ptot) and CANNOT be eliminated.

While the original study needed some revision, the here submitted revised version addresses none of the early design issues. Instead of providing scientifically robust proofs about potential new transient regimes, it further motivates wishful thinking instead of results.

To accept the claims made by the authors about the existence of an intermediate regime leading to flow channelling while transitioning from solitary waves to diapirism, following steps should be included:
1) identification of a physical and testable parameter accounting for focusing
2) systematically testing and reporting of the influence of this physical parameter
3) numerical convergence test to support the robustness of the numerical results (independent of the chosen numerical implementation)

Due to a new procedure in in modeling our solitary waves, we are now able to zoom into the wave, which helps a lot to reach sufficient resolution for solitary waves. We now restrict our models in the paper to models where the compaction length is at least resolved by three grid lengths.

-- Further detailed comments (line numbers refer to the manuscript version 4):
l.9: Not only size but related to compaction length. Size could be kept constant but change in compaction length may lead to similar results

This passage is no longer part of the paper.

l.19-22: No channels will form. Results seem to report a lack of resolution here. To form channels, one needs an asymmetry in compaction versus decompaction rheology. This asymmetry one does not get with the shear rheology. Including porosity dependence in bulk and shear rheology may induce a change in compaction length but no asymmetry.

s.a.

l.45-47: It's the same, as compaction length and radius are interconnected. Changing compaction length using Rt may be the same than changing the bulk to shear viscosity ratio, which will ultimately also impact the compaction length.

Yes, ultimately, we both are changing the compaction length. But in contrast to Scott (1988), we use a porosity dependent viscosity, while he uses a constant viscosity ratio. We changed the sentence to:

Scott (1988) already had a look at a similar scenario. He calculated porosity waves changing the compaction length by altering the constant shear to bulk viscosity ratio, while we want to change the radius of a partially molten perturbation in terms of compaction lengths but keeping the porosity dependent viscosity laws the same

l.68-72: Boussinesq approximation. There is no need for abbreviation since you only use "BA" twice. Also, the wording could be improved here as it is not very clear in the current form.

We got rid of the abbreviation, but we don't know how to improve the wording, as the sentence seams not to be too long and difficult.

---- Section 2.1
Equations (3) and (4) have a pressure issue. How can the same P both be used in the Darcy flow and in the total momentum balance, once relating to fluid density, once relating to total density? Needs revision, modification and clarification.

Equ 10 contains the fluid pressure, indeed, and not the total pressure as claimed by the reviewer. A rigorous derivation of this equation from basic principles can be found in McKenzie (1984, J. Petr. 25, 713 – 765) in Appendix A. In that Appendix equ A9 gives the interphase force and contains the fluid pressure. This interphase force is inserted into equ A7, the momentum equation of the matrix. Inserting also other terms into that equation McKenzie arrives at A16 and furthermore A21 which then is the momentum equation of the mixture (see the mixture density) in the limit of low viscosity fluid (see the deviatoric stress tensor which neglects fluid shear stresses). The pressure in that equation is still the original fluid pressure. McKenzie eliminates the fluid pressure when arriving at A23, and that is what we are doing. In the late 80's many papers used that way eliminating the pressure…

Here from my personal notes on that issue:

It should be noted that the fluid pressure $P$ also occurs in the momentum equation for the mixture, and the intrinsic (averaged) matrix pressure does not explicitly occur. Usually it is different from the fluid pressure (also for the case of neglecting surface tension). If $\nabla \cdot \vec{v}_s = 0$ the intrinsic matrix pressure, the fluid pressure and the effective mixture pressure become equal. In the limit of zero melt porosity the effective mixture pressure and the intrinsic matrix pressure become equal, and the $P$ in equ 4 approaches the intrinsic matrix pressure. This happens smoothly as long as in the limit of vanishing porosity $\nabla \cdot \vec{v}_s$ approaches faster to zero than $\eta_b$ approaches infinity.

Minor notation issue: this section could be enhanced with notation homogenisation. Either adopting the $\nabla$ or $\partial/\partial x$ notation. Also, some i,j,k may be missing if including those.

The $\nabla$ notation is used everywhere but in the momentum equation of the mixture and for the viscous stress tensor. Even though these equations look quite messy in the other notation we think it might be more clear what is done this way.

---- Section 2.3
What linear and nonlinear absolute and or relative tolerances are used (criterion to stop iteration and accept the current solution before starting the next physical time step)?

We think this paragraph is already very detailed and does not need more technical insights. Anyways, we now cite Schmeling et al. (2019) here where the used code is described in more detail.

---- Results Section:
l.233-234: What is observed in Fig.1a-d is simply the evidence of the problem's internal length scale, the compaction length. Although the initial melt anomaly becomes larger, flow still re-organises within a blob of characteristic size given by the compaction length.

Yes, that is true, and we do not state that this is a problem. The shrinking of the waves in comparison to the initial perturbation is totally expected and we just describe this observation. To make it clearer, we added:

This shrinking of the wave in the model is a consequence of reducing the compaction length. The resulting solitary waves have always the same size in terms of $\delta_c$ and become smaller compared to the initial perturbation.

l.244-245: There is no channelling. You may see focusing of melt from an original distribution into a new circular one, but the channels you refer to are numerical artefacts. A model including at least >10 grid points to resolve the channel width will be needed to validate your statement.

s.a.

l.245-267: If you can both model Stokes and porosity waves, why do you need analytical velocity formulation for your lines in Fig. 2? Basically, Fig. 2 should be obtained by tracking your results, or at least some of those. How does one know whether your results match the curves on Fig. 2? One does not need to run any numerical model to reach your conclusions if they're based upon evaluating semi-analytical solutions from literature.

We use these semi analytical solutions from literature, because this solution is for a perfect solitary wave and benchmarked, whose results fit quite nicely to our observations made in this paper. Nevertheless, your point is valid and the reader does not know, whether our waves match these results, but we already compared our waves with the results of Simpson & Spiegelman (2011) in Dohmen et al. (2019). We now state this in the manuscript:

These semi analytical solutions fit quite nicely to our solitary wave models, as already shown in Dohmen et al. (2019).

l.319-325: Be careful with statement. You still do not resolve the small instabilities and those may just be small blobs if properly resolved.

s.a.

l.339-341: Receivable statement upon successful proof that those are not numerical artefacts.

s.a.

---- Numerical issues:
As long as no convergence test neither benchmark (available in e.g., the Appendix of Räss (2019) and Keller (2013)) is provided, the reported results, especially channels, could well be under-resolved and thus numerical artefacts.

We now added the small resolution test to section 2.3., where we can show that the features we observe can still be seen for not sufficiently resolved solitary waves. Anyways, a solitary wave benchmarking is not carried out in this publication, but we carried out a bigger resolution test, and compared our wave solutions to the semi-analytical solutions of Simpson & Spiegelman (2011), which we now cite within the numerical tests in this publication:

The model resolution is a critical parameter in this kind of numerical calculations and should always be kept in mind. With increasing length scale ratio, the compaction length in the model gets smaller and the resolution needs to be increased to keep it equally resolved.

According to several authors (e.g. Räss et al., 2019; Keller et al., 2013) the compaction length should be at least resolved by 4-8 grid points to accurately solve solitary waves. For small length scale ratios this is no problem, where, with a model resolution of $201 \times 201$, up to nearly 30 grid points per compaction length can be achieved. The highest resolution our code can run is $601 \times 601$, which is enough to resolve the compaction length by three grid points for the model with a length scale ratio of 40. Everything above that cannot be sufficiently resolved.

Fig. 1 shows the resulting models for a length scale ratio of 10 for three different resolutions. The pictures were taken after $\varphi_{max}$ has risen approximately 0.25 times the initial Stokes radius ($t' = 0.25$). With increasing resolution, the maximum melt fraction increases strongly from $101 \times 101$ to $401 \times 401$ by approximately 20% but the velocity of $\varphi_{max}$ decreases by 7% (not shown in the figure). Both values converge. Even though the compaction length is not sufficiently resolved in Fig. 1a), one can still observe the main features of the model: A main solitary wave has emerged from the original gaussian perturbation and secondary porosity waves are beginning to emerge within its remains.

The solitary waves modeled with our code have been compared to the semi-analytical solution of Simpson & Spiegelman (2011), and more benchmarking was carried out in Dohmen et al. (2019).

4.1 channelling: As long as bulk rheology is linear; no literature reports any growth of instabilities besides splitting of original wave into new size owing to dynamical change in compaction length. Richardson (1998) shows minor impact of external stresses on blob's shape, but no channelling as such is presented.

s.a.

l.404-405: No channelling here, changes in compaction length will change the characteristic diameter of the spherical wave which needs to be resolved.

s.a.

I.411-414: Unfounded claims. Connolly & Podladchikov (1998) do not suggest following "this upward weakening might not be strong enough to lead to the focusing needed for the nucleation of dykes".

This sentence was deleted, due to the complains above and is not really of importance.

I.422-428: No channelling. As long as there is no asymmetry in viscous compaction versus decompaction, you won't get channels out of blobs. Taking full stress tensor into account and having porosity dependent viscosity will just impact compaction length, nothing else (Räss, 2019).

s.a.

I.439: "the velocities fit quite nicely to the observed model velocities" where does one sees this? You report analytical solution from other authors and your analytical solutions, but nowhere your modelled results. Since your model includes the stress tensor and velocities, it would be very interesting to report those to support your statement and make them receivable.

We now gave the comparison of our results a small paragraph in section 2.3:

The solitary waves modeled with our code have been compared to the semi-analytical solution of Simpson & Spiegelman (2011), and more benchmarking was carried out in Dohmen et al. (2019).

In a single-phase flow case, where the melt is not allowed to move relatively to the solid, the initial perturbation ascends, shortly after beginning, with a velocity of 0.95 times the calculated Stokes velocity, and then slowly decreases as the original Gauss-shaped wave deforms and loses in amplitude.

I. 447-448: A mechanism needs a testable physical parameter, and a verification that this parameter delivers robust and resolution independent results.

s.a.

I. 451: (2) see previous comment.

s.a.

I. 459-463: Good point. Apply it; re-run the suggested simulations with 10 times higher resolutions and longer travel path to convince the reader that you won't get blobs but some real channels being resolved at least with more than 10 grid points.

s.a.

As of the current state, major revisions are warmly suggested.

-- References

Connolly, J. A. D., & Podladchikov, Y. Y. (1998). Compaction-driven fluid flow in viscoelastic rock. Geodinamica Acta, 11(2-3), 55-84.
https://www.tandfonline.com/doi/abs/10.1080/09853111.1998.11105311

Räss, L., Simon, N. S., & Podladchikov, Y. Y. (2018). Spontaneous formation of fluid escape pipes from subsurface reservoirs. Scientific reports, 8(1), 1-11.
https://www.nature.com/articles/s41598-018-29485-5

Räss, L., Duretz, T., & Podladchikov, Y. Y. (2019). Resolving hydromechanical coupling in two and three dimensions: spontaneous channelling of porous fluids owing to decompaction weakening. Geophysical Journal International, 218(3), 1591-1616.
https://academic.oup.com/gji/article/216/1/365/5140152?casa_token=ffbIm7VK8EsAAAAA:L7LuROOcMXTBgosYEdylrae-1rhNCS2E_kVvfn9aOpM3-LnRn5RFtmHEvvFOLvpPlCRssxARrVVa4Zg

Keller, T., May, D. A., & Kaus, B. J. (2013). Numerical modelling of magma dynamics coupled to tectonic deformation of lithosphere and crust. Geophysical Journal International, 195(3), 1406-1442.
https://academic.oup.com/gji/article-abstract/195/3/1406/2874184

Yarushina, V. M., & Podladchikov, Y. Y. (2015). (De) compaction of porous viscoelastoplastic media: Model formulation. Journal of Geophysical Research: Solid Earth, 120(6), 4146-4170.
https://agupubs.onlinelibrary.wiley.com/doi/abs/10.1002/2014JB011258

**Reviewer 3**

**Suggestions for revision or reasons for rejection (will be published if the paper is accepted for final publication)**

This study focusses on the interesting and relevant question of the trade-off between compaction waves and melt-rich diapirs in partially molten systems of the upper mantle and crust. The topic has been covered in the literature in the past but perhaps there would still be room for a study to systematically investigate the transition between the two wellaccepted end-member regimes. However, in my opinion the manuscript in its revised form still suffers from a number of critical flaws that must be addressed fully before publication of the study is advised. The most important issues to be addressed in my view are the following:

- Ratio of length scales as governing model parameter. Previous literature shows clearly that the ratio between the emergent physical length scale, the compaction length (here, "delta"), to the set system length scale (here, "r") is the crucial control on flow regimes between compaction waves and diapirism. If the compaction length is similar or larger than the system length scale, pore fluid segregation is more or similarly rapid than collective flow of both phases as a mixture. Conversely, if the system is much larger than the compaction wave, segregation becomes less relevant and collective flow becomes dominant. Dimensional analysis shows that the square ratio of length scales R = delta^2/r^2 is a dimensionless parameter arising in the governing equations if the system length scale is used to non-dimensionalise length. The parameter in fact arises from taking the ratio of characteristic Darcy and Stokes speeds driven by the buoyancy-contrast between phases (see the recent discussion in Keller & Suckale, GJI, 2019). As this ratio of length scales is at the heart of this study, it is surprising that the authors choose a different, rather more circuitous route in their dimensional analysis of the governing equations. They first introduce the retention number, Rt, not recognising that it is in fact both the ratio of segregation to diapirism speed as well as the square ratio of length scales. Later, apparently as a mere afterthought, the authors bring up the compaction length without putting it into context with their dimensional analysis. They then seek to explain in rather convoluted language how they are varying Rt to obtain an increase in system length scale compared to compaction length. I would regard it as critical to the clarity of their model description and the entire study to instead use a form of dimensional analysis that introduces the governing ratio of length scales clearly from the start and sidesteps the confusing and unnecessary reference to the retention number.

We just never thought of the retention number as a ratio of length scales, but now, as the reviewer stated, we use this ratio as it is much more intuitive than the retention number. We use it in the description of our equations and in the small analysis.

- Numerical benchmarking and resolution testing. One of the main results of this study is the apparent third regime at the transition between compaction waves and diapirism, which the authors characterise as "channelling instability" and discuss in context with rheological melt-shear localisation first introduced by Stevenson (1989). However, as matters stand, the reader can have no confidence that the feature in question is in fact a robust model result rather than a numerical artefact. The numerical setup used in this study is critically flawed. To avoid interactions with boundaries, the domain is 20x larger than the radius of initial perturbations, r. The standard resolution is 200 cells in each direction, meaning r is resolved by 10 grid cells. Unfortunately, the authors then proceed to test parameters where r is roughly equal to or much larger than the compaction length, meaning that in most parameter tests, and notably the ones showing the apparent "channeling instability", the compaction length is not resolved even by one grid step. It would be best practice to present benchmarks of the numerical method against known solutions (or reference published literature if the numerical method has been benchmarked elsewhere). As an absolute minimum requirement, the authors would need to provide a resolution test where it is clear that the solution converges towards a well-resolved geometry, where the compaction length is resolved by at least 8 grid cells. Unfortunately, the authors do not provide either. Nor is it, to my understanding, possible to push the resolution to the necessary levels given the present implementation and model setup.

The reviewer is correct in the statement that the resolution was much too slow to get trustable results. As in the original model setup the required resolution was not reachable, we now changed the model setup a bit. We now let the coordinate system follow the

maximum melt fraction in the model, which allows us to zoom into the initial perturbation, as we no longer need model space for the ascend. Due to that we are now able to reach higher compaction length resolutions. Additionally, a new resolution test shows that the same geometries can be expected, even for lower resolutions.

- Discussion of the apparent "channeling instability". Even if the feature in question be confirmed in well-resolved simulations, its discussion in context of rheological melt-shear channeling remains questionable at best. Previous discussions of channelling instabilities demonstrate that it can arise from strong melt-weakening or non-Newtonian stress-weakening of matrix shear viscosity, decompaction weakening or tensile plastic failure. None of these effects are included in the present model. Therefore it is most likely that the feature in question, if sufficiently well resolved, will turn out to be simply a small compaction wave temporarily escaping ahead of the diapir. However, that would not necessarily be the expected outcome, since the diapir rise speed under the relevant conditions should exceed compaction wave speed. Either way, referring to the feature, if it should persist in revised models, as "channelling instability" is highly misleading.

After changing our model setup and being able to increase the resolution of compaction length we no longer observe channels. Because of that we revised our whole results and discussion chapter.

- Quality of language. The use of language and style in this manuscript falls short of expected standards in international journals. Clear and concise language is important to foster unambiguous udnerstanding. It is highly recommended to have the manuscript edited by a native speaker or professional editing service before resubmission.

Some more detailed comments are given as annotations in the attached PDF file.

L20
The abstract has been rewritten.

L79
The pressure is no declared right after the equation and not later.

L92

L95
Yes, the intrinsic shear viscosity is constant in our models and it is now stated within its declaration.

L102
Thank you for pointing out this mistake. The following equations are based on Sramek et al. (2010) and not Sramek et al. (2007) as falsely stated in our manuscript.

L120
Yes, the pressure was already eliminated, but in another equation. Earlier we use equ. 4 to eliminate the pressure in equ. 3. But now we eliminate the pressure of equ. 4 by taking the curl. This way we do not have the segregation velocities in our equation.

L139
We now use, as stated already above, the ratio of compaction length to model length scale r for the equations and the following analysis.

L154
The ratio was flipped back again.

L167
s.a.

L172
We now use the compaction length earlier in the description of our equations and do not need to attach it at the as an afterthought as stated by the reviewer. Therefore, it is now introduced above at the non-dimensionalization, when it first appears.

L177
Thanks to the reviewers point we do it now as he stated. See above.

L179
Yes, we had concerns about the Stokes flow being affected by the boundaries and we needed some space for the wave to travel to be able to observe the evolution. Periodic boundaries would be a way to tackle this problem but would have been quite hard to implement in our already existing code. We now use a moving coordinate system which basically should lead to similar results and was not too hard to implement. The boundary effects to the Stokes flow get tackled by choosing the pre-factor of the calculated Stokes velocity according to a numerical solution of a cylinder rising within another cylinder and applying it to our square model box. Even though in nature "boundary effects" are never far away, we think it would be best our model as theoretical as possible.

L180
We changed A to phi_max as stated by the reviewer, but we still think the description of the initial perturbation fits best in the model setup. Nevertheless, we now give more information on the perturbation, earlier at the non-dimensionalization.

For $r$ the half width of the prescribed initial perturbation, consisting of a 2D Gaussian bell, is chosen. This is reasonable as the rising velocity in our code is best described by the Stokes velocity, using this radius. The exact shape of the perturbation is given later in the model setup.

L183
s.a.

L189
s.a.

L195
Yes, this is correct, but as stated above periodic boundaries would have been not easy to implement. Anyways, with our prescribed boundary velocities we tackle the problem of melt accumulations at the top satisfactory and do not have any problems. We still need to prescribe them with our new moving coordinate system.

L199
We already thought about mirroring the solitary waves on the boundary, but this would probably lead to more problems. With our old model setup, the observed geometries were about the order of just one grid length, which would have probably led to even more erroneous conclusions. Additionally, even calculating just half of the domain would have been not enough to reach the required resolutions.

L204
The model setup was completely rewritten

L210
s.a.

L212
All used equations were referred to during the description of the code. We think that should be enough.

L213
We now cite Schmeling et al. (2019) where the code was described in detail.

L214
The sentence was removed.

L218
We exchanged the word "damping" by "underrelaxation". We hope that that makes it clearer.

L225
s.a.

L248
s.a.

L251
s.a

L254
s.a.

L293
Sure, we agree. In our code, by default, we solve for advection of composition within the separate phases, and get local evolution of the bulk compositional field. However, in this study we do not distinguish between the chemical composition of melt and solid, melting/solidification is switched off, so we do not have compositional gradients. Pointing this out here may distract the reader.

L318
We now remember the reader that we use a viscosity law that evolves with the melt fraction, while Scott (1988) uses a constant viscosity ratio as model parameter:

This switch from negative to positive mass flux was already observed by Scott (1988), but while he changed the viscosity ratio as an independent constant model parameter, we change the radius and keep the viscosity law the same, still evolving with $\varphi$.

L340
s.a.

L345
The benchmark is now presented in 2.3 Numerical approach

L348
Yes, the resolution test could not reach the resolutions required to optimally resolve our models. With our new model setup we can now reach the required resolutions.

L356
s.a.

L364
s.a.

---

## Author Response (AR3)

I would like to thank the reviewers for their extensive and detailed reviews. Both gave us many suggestions, not only on the scientific part, but also on the written English. I hope that my English will become better in the near future. Anyways, the manuscript was edited, but I had not the time to get professional help. If the manuscript is still not good enough there might be an English editing by the Journal. At least there was one for my last paper published in EGU Solid Earth. If that is not the case, I could find an editing service myself.

In the following you will find the reviewers comments in black and my answers in red. In the marked-up manuscript all changes were also marked in red.

**Reviewer 2**

**Suggestions for revision or reasons for rejection (will be published if the paper is accepted for final publication)**

The submitted revised version of the revised manuscript by Dohmen and Schmeling systematically investigates magma ascent dynamics in order to capture the transition from the solitary wave regime to diapirism. The authors explore this transition by varying the relative compaction length of the system - here by changing the model extent while keeping the compaction length constant.

In this latest revision the authors finally address the study's major early design issue. The current version is scientifically sound and brings some insight on the transition without extensively reporting numerical artefacts.

The current manuscript lacks however in quality regarding the writing style which significantly affects readability. You'll find hereafter some ideas on how to improve it. Another solution would be to get the manuscript revised for language by a professional language service or get some advice from a native speaker.

The author should clarify the pressure issue and elimination discussed between equations (3)-(10). I agree it is not the scope of the paper to sort out the confusion and extra complexity introduced by McKenzie (1984), but the current explanation is not consistent. It would be of great help to the reader to have well defined concepts with clear names referring to that are kept constant throughout the entire manuscript. To this end, it would be appreciated to have a clear definition of what is the melt, the fluid, the effective properties, the mixture and other quantities introduced on a non-systematic basis throughout the paper. Then, once this framework and naming conventions are set, define the pressure as it should. I suspect in the current form the pressure called fluid pressure (and missing the subscript f) may not be the pressure of the pore-fluid but a composite fluid pressure defined by McKenzie.

The pressure in our equations is the fluid pressure, where we now added a subscript f. The fluid pressure is balanced by the lithostatic pressure for solid density, the compaction pressure and the dynamic pressure (eq. 5). The dynamic and compaction pressure are both included in the stress tensor (eq. 6), as we state now in its definition. The fluid pressure in our equations should therefore be the fluid pressure and not a composite fluid pressure. I agree that in our description the pressures are not defined individually, which may lead to confusions, but the description should be consistent. The pressure in eq. (3) is the fluid pressure as well and we can use eq. (5) + (6) to achieve eq. (10).
I cannot find any wrong notations in our description, but we now use $\bar{\rho}$ for the mixture density and $P_f$ for the fluid pressure, which should clarify some possible confusions. Additionally, the volume viscosity is now given by $\zeta$, the shear viscosity by $\eta$ and the dynamic fluid viscosity by $\mu$, because we changed the term bulk viscosity to volume viscosity and adapted our notation.

Based on these considerations, I would recommend this draft for publication after extensive minor revisions (more than just fixing the detailed comments below).

**General comments**
Hereafter some ideas and suggestions on how to enhance the writing style applicable to the entire manuscript. Sentences should be kept short and as concise as possible. One way of achieving this is to switch to the active voice wherever it is possible.

Action verbs may also help getting straight to the point; consider using more precise verbs rather than "doing".

The text is populated with too much useless linking words. One possibility to fix this would be to

remove all parasite linking words, read through the text, and add them back in places where having them missing would seriously alter the meaning or the logical reasoning.

The text currently includes a lot of familiar "spoken" jargon. Consider removing these or replacing them with more scientific and precise concepts.

Regarding the layout and style, it is considered good practice to have inline equations in the text, preferring horizontal fraction expressions to / for better readability. Also, referring to figures and equations in a passive fashion (e.g. "Our results confirm A (Fig.X)" instead of "Figure X shows that our results confirm A."") may often help to increase flow and readability.

**Specific comments**
l.6 Consider capitalising the "Earth" throughout the text

"Earth" has been capitalized throughout the text.

l.8 solitary porosity waves -> Solitary waves of porosity

I think the term "solitary porosity waves" is more common and is not necessarily less clear. Nothing has been changed.

l.10-11 a mechanism could be dominant, not really a wave or a diapir. Consider a more accurate formulation.

The sentence has been adjusted to:

Thus, the size of a partially molten perturbation in terms of compaction length controls whether material is dominantly transported by porosity waves or by diapirism.

l.16 For enhanced readability I would suggest a full stop after "flux" and starting a new sentence for the melt.

The suggestion was accepted:
If the perturbation is of the order of a few compaction lengths, a single solitary wave will emerge, either with a positive or negative vertical matrix flux. If melt is not allowed to move separately to the matrix a diapir will emerge.

l.37-38 "However, ..." consider simplifying the sentence making it straight to the point.

Regarding suggestions made in the Introduction:
The introduction was to large parts rewritten, following most of the suggestions made by the reviewer.

l.43 "On the other hand" suggests "on the one hand" to appear first. Consider removing unnecessary linking words for enhanced clarity.

l.48 Consider being more precise than "transition looks like" which does not sound very scientific. What interests you in the transition regime ?

l.51 Consider being more factual in this last sentence switching subject from authors to end-member scenarios.

l.54 please refine or clarify the concept of "partially molten scenario".

l.56 "and look especially [at] what happens" doesn't sound very scientific. Consider being more precise and avoid using familiar jargon in a scientific publication.

l.62 Be more specific about "more viscous" -> I suspect you mean feature larger viscosity values, e.g.

l.64-65 "Based on ..." sentence's construction is suboptimal making it hard to get the point. Consider revision for more clarity.

l.66 Consider adding "single phase" to "the Stokes equations" ?

l.72-74 "In the present Boussinesq ..." please rephrase this sentence making the comparison clear.

We already revised the part about the Boussinesq approximation and explained it quite detailed. I don't know how to explain it better, but I think it should be already clear enough.

Section 2.1 Revise usage of fluid and melt. The terminology is mixed up and maybe miss-used in some places in this section, giving the reader hard time to follow the mathematical model formulation.

I could not find any miss-used subscripts. Maybe some irritations came along with the density, where we used $\rho$ for the mixture density. We now use $\bar{\rho}$ as it is most common.

l.84 If P is the fluid pressure, consider adding the subscript f to it for consistency.

The subscript f was added to all P throughout the text.

l.85 It is unclear how P, being apparently Pf, would actually include a lithostatic component.

s.a.

l.86 What is P in the Stokes equation for the mixture, Pf or another pressure ? If P stands for Pf here, it is confusing as this would mean fluid pressure balances the mixture buoyancy forces and shear stresses.

The second term of the stress tensor is the compaction pressure, which we now explicitly state in the definition of the tensor:

… and $\boldsymbol{\tau}$ is the effective viscous stress tensor of the matrix including both shear and compaction components…

l.88 Eq. (5) is known as Kozeny-Carman relation. Consider stating it and referring to e.g. Costa 2006: Costa, A., 2006. Permeability-porosity relationship: a reexamination of the Kozeny-Carman equation based on a fractal pore-space geometry assumption, Geophys. Res. Lett., 33(2), L02318.

The Kozeny-Carman relation is now mentioned and cited:
This relation is known as the Kozeny-Carman relation (e.g. Costa, 2006).

l.88 power-law "n" definition seems missing

Yes, it was missing and was now added:
with $n$ being the power-law exponent constant, usually equal to 2 or 3.

l.89 ηf, g and ρ refer to Eq.(4), consider moving these definitions right after the equation they appear in.

The equations and the definitions of the parameters have been slightly rearranged:
The momentum equations are given as a generalized Darcy equation for the fluid separation flow

$$\vec{v}_f - \vec{v}_s = -\frac{k_\varphi}{\eta_f \, \varphi} \left( \vec{\nabla} P_f - \rho_f \vec{g} \right), \tag{3}$$

where $\rho_f$ is the fluid density and $P_f$ is the fluid pressure (including the lithostatic pressure), whose gradient is driving the motion. $k_\varphi$ is the permeability that depends on the rock porosity

$$k_\varphi = k_0\varphi^n, \tag{4}$$

with $n$ being the power-law exponent constant, usually equal to 2 or 3. $\eta_f$ is the melt dynamic viscosity, $\vec{g}$ is the gravitational acceleration. The Stokes equation for the mixture is given as

$$\rho\vec{g} - \vec{\nabla}P_f + \frac{\partial\tau_{ij}}{\partial x_j} = 0. \tag{5}$$

$\rho$ is the density of the melt – solid mixture and $\tau_{ij}$ is the viscous stress tensor…

l.90 τij is the mixture, solid, total, melt stress tensor ? Please be more precise.

The description of the tensor is now more precise:
… and $\tau$ is the effective viscous stress tensor of the matrix including both shear and compaction components…

l.94 "simple" does not provide any further information - consider removing it.

"simple" was removed.

l.119-121 Be consistent with the comparison 1 Pa.s - 1e14 Pa.s and then you switch 1e20 Pa.s - 1e16 Pa.s; consider choosing increasing or decreasing values for both.

The values now increase for both cases.

l.155 Consider rephrasing this sentence to the active voice for more clarity.

The sentence was rephrased:
All quantities in the other equations are simply replaced by their non-dimensional primed equivalents (eqs. (1), (2), (6), (11), (12), (13), and (14a)).

l.157 "We can now compare…"

The sentence was changed according to the reviewers suggestion.

l.157-159 Long and complicated sentence to state something trivial. Consider revising for enhanced clarity.

I don't think this sentence is too long and complicated, but we changed the sentence a bit and is now hopefully more clear:
We now compare the two limits, where segregation or two-phase flow dominates (solitary wave regime), and where fluid and solid rise together with the same velocity as partially molten bodies, which we identify with the diapir regime.

l.159 Consider replacing familiar expressions with more accurate terms. E.g. "this can be done" -> "we compare the characteristics…"

The suggestion was accepted.

l.168 Why to have italic text here. Consider clearing formatting or implementing corresponding format to the "diapir limit" text.

The italic formation was removed.

l.174 Complete "equ" word

The word is now complete

l.190 Consider replacing big by large and spell out what you mean with "the other way around". This sentence could be formulated as a dual comparison in a concise manner.

The sentence was rewritten:
Smaller amplitudes lead to a switch at a smaller radius and larger amplitudes to a switch at a larger radius.

l.194 Is "Gaussian wave" the most suited terminology for describing your initial condition ? Consider revising it, maybe a "Gaussian profile" or distribution ?

"Gaussian wave" was replaced by " Gaussian bell-shaped".

l.199-200 What do you refer to as "model series" ? What do you refer to as "parameter range towards the diapiric regime" ?

The sentence was revised:
In our model series we vary the ratio of Stokes radius to compaction length from 1.8 to 48 to explore the transition from solitary wave towards diapiric regime.

l.200-201 what unit is your 201 x 201 model resolution? I suspect it is the number of grid points. Please add it. What does a "high length-scale ratio" stand for ? Consider being more precise.

The sentence was revised:
The resolution of the models is chosen to be at least $201 \times 201$ grid points and was increased for higher ratios of Stokes radius to compaction length so that the compaction length is resolved by at least 3-4 grid points.

l.203 "at the top and the bottom" of what ? What about "At the top and bottom domain boundaries we ..."

The suggestion was accepted.

l.203-205 Be more specific and scientific in the description. Simply state what boundary conditions you implement and for which reason in an active voice.

The paragraph was revised:
At the top and the bottom domain boundaries, we prescribe an out- and inflow for both melt and solid, respectively, to prevent melt accumulations at the top. The segregation velocity of the background porosity $\varphi_0$ is calculated using equation (17) without the viscous stress term. The corresponding matrix velocity is calculated using the conservation of mass.

l.208-209 What does the reader get as add-on value knowing that the lateral BCs are "mirroring" ? Why don't you just state "We enforce no flux horizontal boundary conditions." ?

The suggestion was accepted.

l.209 The power-law exponent should be defined in Section 2.1 and n=3 could potentially show up there already if it is constant throughout the study.

The power-law exponent is now defined in Section 2.1, but I think it is still important to have it here in the Model description.

l.210-214 You implement a moving frame. Could you be more precise or drop the shifting

information ? Once φmax reaches L/2+dz you shift the entire model down from one dz. But then you repeat this for every next dz travelled by φmax ?

Yes, the information on how we realize this moving coordinate system might be not so important. The paragraph was revised:
To run models for a longer, practically infinite, amount of time we let the models coordinate system follow the maximum melt fraction. This procedure allows us to zoom into the perturbation and follow it, not knowing its velocity and without carrying out any interpolations, which would strongly influence the model.

l.217-218 What about "We discretise the non-dimensional set of equations (xx-yy) using finite-differences on a staggered? regular? grid and solve the system using the FDCON code (Schmeling et al., 2019)"?

The sentence was rewritten:
We discretize the set of equations using finite differences on a staggered grid and solve the system using the code FDCON (Schmeling et al., 2019).

l.250-254 It is motivating to read that benchmarking was carried out. It would be highly appreciated if the statements could be supported by one or two figures in e.g. an Appendix.

I think adding an Appendix just for a figure of a numerical test is not necessary, as Fig. 2 already shows the important outcome of the numerical test, namely the evolution of the initial perturbation with the resolution used for model series is approximately equal for higher resolutions. A paragraph without a figure about the convergence of velocity and melt fraction amplitude should be enough.

l.257-259 This explanation is very helpful to remind the reader about the experiment design.

l.267 I would reformulate the sentence making such as "We compare the observed solitary wave velocities (Fig3b-e) to equivalent Stokes velocities for a diapir based on eq. (15)". Making more concise sentences without decoration words will strengthen your sayings and clarify the text a lot.

The suggestion was accepted.

l.273-274 "fits quite nicely". Either it fits, or not. If it fits not exactly, select a precise word instead of "quite nice".

The sentence was revised:
These semi analytical solutions are in good agreement to our solitary wave models, as already shown in Dohmen et al. (2019).

l.274-275 Here and in some other paragraphs, avoid using "can". It reads you are not trusting your results or awaiting the reader to approve your claims. It does not read well.

This sentence was changed. And we tried to avoid "can" in other sentences as well:
The velocities in this figure correspond to ratios of solitary wave velocity to initial perturbation Stokes velocity.

l.315-317 Consider switching this sentence to the active voice and making a linear construct, subject, action verb, complement. Also consider assessing whether "Just in the case" is really needed.

The paragraph was revised:
A classical diapir will evolve only in cases with zero compaction length ($r = \infty \cdot \delta_c$), i.e., melt is not able to move w.r.t. the matrix (Fig. 3k). Here, no focusing into solitary waves can be observed and transition time is infinity.

l.345 Consider switching to the active and affirmative voice. Report what you did and not what you think you may have done, e.g. "A quantitative analysis of ...".

The sentence was changed:
The transition from solitary waves towards diapirism on qualitative model observations was so far only based on observations.

l.349-351 "soli" and "dia" not being variables, consider a non-italic font style.

"soli" and "dia" are no longer in italic.

l.369 Scott isn't himself "in the two-phase flow regime". His results suggest it. Consider modifying here and other places in the text where you refer to authors instead of author's results.

The sentence was revised:
Both describe the transition from a two-phase limit towards the Stokes limit, but in our formulation, we are able to reach the Stokes limit while Scott's formulation (1988) is restricted to two-phase flow.

l.400 "that rise diapiric as a swarm" -> "that rise as a diapiric swarm".

The suggestion was accepted.

l.410-415 The work of Keller 2013 may provide some references to you claims

The paragraph was changed and Keller et al. 2013 was cited and discussed shortly above.

l.416-421 Unclear formulation and complicated sentence construct. Consider revising and simplifying.

The paragraph was rewritten:
Even though most diapirs should, according to our models, disintegrate into numerous solitary waves, not all will inevitably. Within regime (1) solitary waves are possible and most probably expected but the deeper we are in regime (2) the less expected is the disintegration because a long time is needed to build up. In nature, different from our models, they cannot rise for an infinite amount of time.

l.436 Consider changing the title "Other issues" to "Model limitations" or another positively relevant item. "Other issues" sounds like previous material is already an issue which is not the case.

The suggestion was accepted and the subsection is now called "Model limitations".

l.449 Although solitary waves may be observed on non-resolved resolutions, their properties such as velocity would be non accurate.

l.464 What does a and b stand for or refer to ? Please add precision there.

The explanation for a and b is written in the next sentence. To make it clearer a and b was added to the number of the enumeration:
(1a + b) Solitary wave a and b, (2) solitary wave composite diapirism and (3) diapirism.

l.472-473 "might be no longer resolved properly" -> "may not be under-resolved to allow for ..."

The suggestion was accepted.
* * *
Figure captions:

Fig.2:
"The six panels depict..."
"resolutions" -> "numerical (grid) resolution"

The suggestions were accepted.

Fig.3:
A general statement of what the figure is about is missing, e.g. "Melt ascent morphology as function of compaction length".
"a) Initial conditions of the model valid for all cases apart of the change in compaction length".
"b-j) Melt fraction distribution after ... for length scale ratios varying between 2.4 and 48."
"k) Diapiric rise resulting from a compaction length of zero."
"l) Models' transition time as function of length scale ratios varying between 1.8 and 120. ..."
The figure caption was adjusted according to the reviewers suggestions:
Fig. 3: Melt ascent morphology as function of initial perturbation radius in terms of compaction length.

a) Initial conditions of the model valid for all cases apart of the change in compaction length. b-j) Melt

fraction distribution after $t' = 0.2$ for length scale ratios varying between 2.4 and 48. k) Diapiric rise

resulting from a compaction length of zero at $t' = 9$. l) Models' transition time as function of length

scale ratios varying between 1.8 and 120. The transition time gives the time after which the main wave

has reached a solitary wave status.

Fig.4:
Consider replacing "show" by "refer to"

"show" was replaced by "refer to".

Fig.5:
"The upper row panels depict..."
"... from left to right, respectively"
"The bottom row panels depict the corresponding ..."

The suggestions were built in:
Fig. 5: The upper row panels depict the solid and fluid mass fluxes of a horizontal line cutting through

the maximum melt fraction at timesteps where the main wave has just reached the status of a solitary

wave. These timesteps are $t' = 0.02; 0.068; 0.155; 0.416$ from left to right, respectively. The bottom

row panels depict the corresponding melt porosity fields. All quantities shown are non-dimensional.

Fig.6:
A general statement of what the figure is about is missing.
"a) Solitary wave (blue) and ..."
"The dashed lines highlight the transition in the regimes."
"b) Ratio of maximum... in the entire model."

The suggestions were built in:
Quantitative parameters as function of initial perturbation radius in terms of compaction length. a)

Solitary wave (blue) and diapir (red) partition coefficients for several initial perturbation radii. b) Ratio

of maximum fluid velocity to maximum absolute solid velocity in the entire model.

**Reviewer 3**

**Suggestions for revision or reasons for rejection (will be published if the paper is accepted for final publication)**
I would like to thank the authors for thoroughly addressing the issues raised in the previous round of revisions. The revised work now clearly demonstrates that the numerical method produces valid and well-resolved results, introduces a more appropriate dimensional analysis to explain the dominant process regimes, and (mostly) avoids interpretations not clearly supported by the model results. As such, the manuscript is now close to a publishable state.

However, a few issues remain to be addressed.

One of the main concern at this stage is the quality and conciseness of language used throughout. I'm afraid, the text is currently of a quality and style of writing not suitable to publication in an international journal. The instances of incorrect grammar, convoluted and repetitive descriptions, and informal or otherwise inappropriate expressions are too many to usefully point out in detail. I strongly urge the authors to apply thorough copy-editing throughout and in places substantial rewriting to the text.

While the early introduction of the concept of compaction versus diapir length scales and segregation versus collective flow speeds is very welcome, I recommend a general rewrite of the Introduction where the broader context and the specific question or hypothesis of the research are more clearly explained, and where the end-member regimes and their contrasting length and speed scales are more systematically introduced and discussed in light of the literature. At present, the text in some places appears incomplete (e.g., no mention that the research is motivated by partial melt transport in the asthenosphere) and in other places disjointed or oddly sequenced.

I recommend the authors consider condensing some of the lengthy and at times repetitive descriptions and discussion of the results. In a number of instances, the authors repetitively point out features that are clearly visible in the figures without adding more information to the discourse. The clarity of characterisation of regimes between compaction wave- and diapir-dominated is obfuscated by repetitive discussion and unclear or inconsistent terminology. Similar arguments are stated multiple times in different contexts, but no clear and consistent terminology is introduced to characterise the regimes or the main metrics used to characterise them. For example, in one instance the authors refer to the diapirism regime as "the solitary wave composed diapiric uprise regime". I believe it should be possible to rewrite the Results section to condense and clarify the observed model behaviour and the following regime classification using concise descriptions and clear and consistent terminology. It would allow the interesting results to be understood more clearly.

Finally, in the Discussion the authors attempt to argue that larger diapirs will generally break up into trains of smaller compaction waves, and that these could be expressed in pulsating volcanic activity. I'm afraid I find this line of argumentation as it is currently stands entirely implausible. Firstly, the authors fail to clearly discuss that their analysis only applies to porous flow of melt within generally partially molten domains at low melt fraction. Previous work has shown that considering more realistic rheologies (e.g., exponential melt-weakening, non-Newtonian or visco-plastic matrix viscosity, decompaction weakening) all lead to substantial modification of melt transport such that the classical picture of compaction waves versus melt diapirs might not apply at all, or at least not everywhere in the partially molten mantle. These limitations should be clearly pointed out and taken into account when discussing implications. Secondly, there are a number of complex processes at play in transferring melt arriving from mantle melt source into the lower lithosphere to volcanic centres at the surface. The ones that come to my mind are subject to their own time and length scales which will likely buffer out or entirely overprint any periodicity of melt transport in the mantle. In my view it is therefore not plausible that the model results presented here have a direct bearing on observed volcanism at the surface. They may still be relevant for some partially molten domains in the mantle or ductile crust, and that is where I would see the main contribution of this work to our present understanding.

The criticism about this paragraph in the discussion is justified and we therefore dropped it.

I further point out a number of minor suggested clarifications and corrections in the annotated manuscript attached to this report.

Tobias Keller (Tobias.Keller@glasgow.ac.uk)

Introduction

The introduction was to large parts rewritten, following most of the suggestions the reviewer made.

L67

The headline was changed to „Methods".

L74-76

The suggestions of the reviewer were accepted.

L85

The suggestion was accepted.

L86

The notation was adapted and is now following the reviewer's suggestion.

L89

The mixture density is now given by $\bar{\rho}$.

L90

The full definition of the viscous stress tensor as given by the reviewer is now given.

L91

s.a.

L92

Yes, we agree that "bulk viscosity" might not be the best name. But we do not fully agree that compaction viscosity is the best name, so instead we will us "volume viscosity", which should be good name. The term "compaction viscosity" implicates that this viscosity solely affects the compaction, but it is also affected by the shear viscosity.

L151

The definition of the compaction length of the reviewer was accepted.

L158

The term "batch melting" was abandoned.

L164

Yes, omitting the geometric prefactors would make this analysis simpler, but later in the analysis when the switch is calculated these play a role. With prefactors the switch is at a ratio of 48, without at 28. The lines in Fig. 1 get noticeably shifted to the left and no longer reflect the results of our models so nicely. We therefore keep the prefactors.

L164

Thank you for pointing this out. Due to that I could eliminate a typo in this equation. $\frac{v_{sgr}}{v_{st}} \sim \frac{\delta_c^2}{r^2}$ and not the other way around, as stated before. Furthermore, the compaction length is in fact the constant compaction length for the background porosity of our model. Therefore, we should notate that and use $\delta_{c0}$. The typo was eliminated, and the compaction length was correctly notated.

L204

This part of the sentence was erased as it does not help the reader to understand the model setup.

L212

The part about how the tracking coordinate system is carried out was erased due to simplifications stated by the other reviewer. The part about the zooming was referring to our earlier model setup where the perturbation was smaller in the model box. It would most certainly irritate the reader and was erased anyways.

L239

The suggestion was accepted.

L243

It was added that they converge for resolutions higher than 51x51.

L246

The suggestion was accepted.

L273

The sentence was rewritten, following the suggestions of the reviewer:

These semi analytical solutions are in good agreement to our solitary wave models and differ only by 3-5% percent in velocity, as already shown in Dohmen et al. (2019).

L286

The misspelling was corrected.

L295

The misspelling was corrected.

L318

The suggestion was accepted.

L328

The suggestion was accepted.

L343

The misspelling was corrected.

L355

The sentence was rewritten so that it now says that the sign changes, which corresponds to upward flowing matrix:

With increasing radius $C_{\mathrm{dia}}$ increases until it changes its sign, and the matrix flows upward, at $r \approx 20 \cdot \delta_c$. It eventually becomes bigger than $C_{\mathrm{soli}}$ at $r = 36 \cdot \delta_c$ and then approaches 1 for bigger radii.

L366

"He" was replaced by "they".

L371

The suggestion was accepted.

L380

The suggestion of the reviewer was accepted and the second regime is now just call "diapirism-dominated regime".

L385

The third regime was replaced by the endmember of the second regime.

L388

We now state that this is only true in the present model and cite Keller et al. (2013) to give an exception:

In every other case, in the present model, where fluid is able to move w.r.t. the solid, at some point all diapirs will evolve into a swarm of solitary waves which can be infinitely small compared to the initial perturbation. However, this is expected to happen only after a long distance of diapiric rise. In cases where the size of solitary waves is comparable to the perturbation (e.g. regime (1)) this will occur sooner and in cases, where solitary waves are much smaller, later. Their observation is mostly limited by resolution. For models that allow for the diapir to grow (e.g. Keller et al., 2013) they may not dissolve into solitary waves, as it approaches the single-phase limit.

L411

The concerns of the reviewer are correct. The scenario of a magma chamber was abandoned and replaced by "a partially molten region within the mantle".

L427+428

s.a.

L436

We changed this subsection a bit and changed its title to "model limitations". We added a small discussion about the internal circulation of diapirs smearing out the emergence of solitary waves and connected the paragraphs to allow for some kind of "story". The subsection was kept.